# CROSS-DOMAIN PRE-TRAINING OF TRANSFORMERS ON TEXT-ATTRIBUTED GRAPHS VIA RANDOM WALKS

## ABSTRACT

Pre-training large-scale models with diverse data using the Transformer architecture has driven significant advances in natural language understanding. Motivated by this success, we explore pre-training strategies for graph representation learning that leverage the flexibility of Transformers. A key challenge is enabling a sequence-based Transformer to effectively encode graphs of varying sizes and from diverse domains. To address this challenge, we represent nodes as collections of random walks, allowing the Transformer to learn node embeddings from sequential contexts. We provide theoretical analysis on the expressive capacity of this representation for distinguishing graph structures. We also introduce a novel context prediction loss tailored to random walks. Empirically, we show that the proposed pre-training strategy can be adapted to various downstream graph tasks, highlighting its promise for processing and reasoning with graph-structured data.

## 1 INTRODUCTION

Recent progress in natural language processing shows that large Transformer models, when trained on diverse data and properly adapted, can generalize to a wide range of tasks (Bommasani et al., 2021; Vaswani et al., 2017). Moreover, when scaled with sufficient parameters, they elicit emergent abilities that are not present in smaller models (Wei et al., 2022). A key to sustaining the performance of these Transformers is the concurrent scaling of the training data and the model size (Kaplan et al., 2020; Hoffmann et al., 2022). These insights motivate the question of whether a similar pre-training paradigm can apply to graphs (Mao et al., 2024), where a model is pre-trained with and adapted to various kinds of graphs.

Compared to text, graphs introduce unique challenges for designing Transformer-based models. First, graph data are inherently non-sequential. To achieve invariance or equivariance to node permutations, one cannot rely on standard positional encodings as in sequence models; instead, the graph structure must be incorporated through mechanisms such as structural encodings or attention biases. Second, graph sizes vary significantly, ranging from fewer than ten nodes to billions in practice. This variability causes difficulties in batching graphs. For small graphs, nodes can be chained together to form a sequence, but large graphs will need to be partitioned into subgraphs, which may break connections between different partitions.

Our objective is to design a general methodology toward pre-training a Transformer model for various graphs, with the following desiderata:

**D1:** The model should be pre-trained across diverse graph datasets in a self-supervised manner, without reliance on task-specific labels.

**D2:** The pre-trained representation should be adaptable to a wide range of downstream tasks, supporting transfer to graphs from new domains.

**D3:** The model should flexibly handle graphs with different structures and scales, from small molecules to large citation networks.

**D4:** The architecture should be able to capture long-range interactions when they are relevant to the target task.

These desiderata require a holistic design of the model architecture, input and output formats, training objectives, and downstream adaptation strategies. We begin with interrogating the strengths

and limitations of the two most widely used graph deep learning architectures: Graph Neural Networks (GNNs) (Li et al., 2016; Kipf & Welling, 2017; Hamilton et al., 2017; Gilmer et al., 2017; Veličković et al., 2018; Xu et al., 2019; Gasteiger et al., 2019; Chen et al., 2020) and Graph Transformers (GTs) (Dwivedi & Bresson, 2021; Kreuzer et al., 2021; Ying et al., 2021; Wu et al., 2021; Rampášek et al., 2022; Chen et al., 2022; Ma et al., 2023). The computation pattern of GNNs is neighborhood aggregation; as a result, the main challenge is a uniform network depth for all graphs and the handling of long-range interactions if exist (Dwivedi et al., 2022). Experience suggests that GNNs for different domains vary substantially in depth. While one may attempt to take the maximum depth, which also resolves the long-range challenge, on other occasions deep GNNs suffer from the over-smoothing problem (Li et al., 2018). Mitigation approaches exist, such as residual connections (Chen et al., 2020) and edge removals (Rong et al., 2020), but many layers aggravate the neighborhood explosion problem because typical neighborhood sampling methods (Hamilton et al., 2017; Chen et al., 2018) will still create an enormous neighborhood. Recent approaches (Liu et al., 2024a; Wang et al., 2024) sample small-hop neighborhoods for large graphs so that the GNN depth can be more flexible, but these neighborhoods still miss long-range information. On the other hand, GTs are a principled approach to incorporating this information because of the pairwise attention, but this comes at a computational cost: for a graph with $n$ nodes, it typically takes $\mathcal{O}(n^2)$ time to compute the attention scores. Much effort has been devoted to scaling GTs to large $n$, such as (i) using kernel approximation of the softmax attention (Wu et al., 2022; Choromanski et al., 2021), (ii) taking a hierarchical approach (Zhu et al., 2023), and (iii) changing the input from a sequence of all graph nodes to a sequence of sampled neighbors of one node (Zhao et al., 2021; Zhang et al., 2022; Chen et al., 2023a). However, approaches (i) and (ii) still have trouble with batching when graphs have varying sizes and approach (iii) weakens the incorporation of long-range information.

In this work, we propose the Random Walk-Based Pre-Trained Transformer (RWPT). The main idea behind this model is the use of multiple random walks to represent one node and to leverage a Transformer backbone for flexible sequence modeling. RWPT differs from usual GTs in that the Transformer input is neither the whole graph nor a sequence of sampled neighbors. Instead, multiple random walks are taken from a root node, forming ordered sequences including near neighbors and faraway nodes. Random walks are a revival of the early node embedding methods prior to GNNs, such as DeepWalk (Perozzi et al., 2014) and node2vec (Grover & Leskovec, 2016), which permit favoring depth in addition to breadth when considering node co-occurrences. They are key to our holistic design that meets the four aforementioned desiderata: Random walks resolve the batching problem of GTs when training graphs have drastically different sizes (**D3**); they encode a larger receptive field and better cope with long-range interactions (Chen et al., 2024a), compared with small-hop neighborhood sampling (**D4**); and they allow the pre-training with any cumulation of graph datasets (**D1**) as well as the separation of self-supervised pre-training and downstream adaption (**D2**). In addition, we show that combining single-source random walks with shortest-path distance positional encodings leads to expressive node representations capable of reconstructing and distinguishing balls (the ego-graph of a node induced by its $r$-hop neighborhood). Therefore, it forms a suitable sequential format for node representation learning.

Our contributions are as follows:

- We outline four key desiderata for pre-training a Transformer model across diverse graphs. The first two are analogous to objectives in natural language models while the other two are tailored to graph-structured data.

- We propose RWPT, a Transformer framework that meets these requirements and addresses the limitations of current graph deep learning models (GNNs and GTs). Central to RWPT is the use of multiple random walks to represent a node, which subsequently invokes the accompanying designs of positional encoding, attention masking, and training loss for the Transformer.

- We provide a theoretical analysis of the proposed design of sequential representation for nodes, establishing its expressivity in distinguishing local neighborhoods and supporting its suitability for representation learning.

- We conduct comprehensive experiments to demonstrate the effectiveness of RWPT compared with (semi-)supervised and self-supervised methods, highlighting its transferability and adaptivity in cross-domain and cross-task uses.

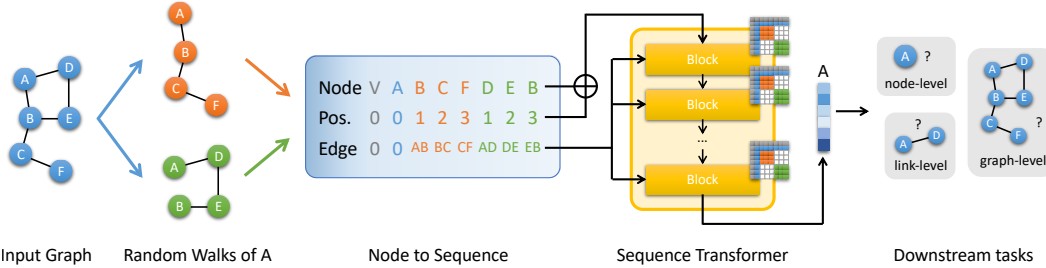

Figure 1: Pipeline of RWPT. Each node is represented by multiple random walks formulated into one positionally encoded sequence, augmented with domain information. The sequence is processed by a Transformer with a per-walk attention mask. The node representation is extracted from the output. The model is fine-tuned through training a prediction head dedicated to the downstream task.

## 2 RELATED WORK

This work builds on recent progress in graph representation learning and unified models. While large language models (LLMs) have driven the development of general-purpose architectures in natural language processing, similar efforts to design unified models for graphs are emerging. Our approach is most relevant to two recent methods: OFA (Liu et al., 2024a) and GFT (Wang et al., 2024).

OFA (Liu et al., 2024a) trains a single GNN across multiple datasets and tasks in a supervised fashion. For node- and link-level tasks, the model operates on $\ell$-hop neighborhood subgraphs rather than the entire graph. However, it remains unclear whether this framework can be adapted for pre-training without task-specific supervision. GFT (Wang et al., 2024) pre-trains a vocabulary of embedding vectors derived from quantized $\ell$-hop neighborhoods, which are encoded using a GNN. During inference, it assigns the nearest pre-trained embedding to a new neighborhood subgraph for downstream tasks. Its reliance on small $\ell$-hop neighborhoods limits its ability to capture long-range dependencies. To address this, we utilize random walks that can reach faraway contexts.

A comprehensive overview of related work—including GTs, pre-trained GNNs, LLM-enhanced graph methods—is reviewed in Section A.

## 3 METHODOLOGY

In this section, we elaborate on the details of the proposed RWPT model. It is a holistic design that involves not only the neural architecture but also the data representation and training. Three aspects are highlighted: the formulation and encoding of the input sequence, the attention mechanism, and the pre-training loss. Figure 1 shows the feedforward flow of RWPT during inference.

### 3.1 RANDOM-WALK REPRESENTATION OF NODES

Random walks form ordered sequences, which are natural inputs to the Transformer. In action, assume that $i = i_0$ is the node of interest (i.e., the root node). We run $k$ independent random walks of length $\ell$ starting from $i_0$ and denote by $i_s^r$ the node at step $s = 1, \ldots, \ell$ and walk index $r = 1, \ldots, k$. We concatenate all nodes, walk by walk, to form a sequence

$$\text{seq}(i) = [i_0, i_1^1, \ldots, i_\ell^1, i_1^2, \ldots, i_\ell^2, \ldots, i_1^k, \ldots, i_\ell^k]. \tag{1}$$

One may interpret the union of the random walks to form a sampling of the large-hop neighborhood of $i$. This sampled neighborhood differs from a common one resulting from neighborhood sampling (Hamilton et al., 2017; Chen et al., 2018) in that a larger walk length $\ell$ is permissible but the number of hops in neighborhood sampling is limited, because the number of sampled nodes is exponential in the hop count. Note that a node in $\text{seq}(i)$ may appear multiple times (because it is sampled by different walks), but its embedding in different appearances may be different because of edge features.

Since we intend to pre-train the Transformer with multiple datasets, to distinguish graphs from different domains, we introduce a virtual token $v$ for each dataset. We prepend this token to the

sequence (1); that is, the full sequence for a node $i$ input to RWPT is

$$s(i) = [v, \mathrm{seq}(i)]. \tag{2}$$

## 3.2 SEQUENCE ENCODING

Like a standard Transformer whose input sequence is encoded by token embedding and positional encoding, we define our input node features and positional encoding for RWPT. Additionally, we incorporate edge features by formulating them into the sequence and adding them to the input of each Transformer block.

**Unified input node features.** One of the technical difficulties in unifying graphs from different domains is the varying node features and dimensions. LLM offers a perfect mechanism to mitigate this difficulty (Chen et al., 2023b; He et al., 2023a; Liu et al., 2024a). Nearly all graphs from practical problems are equipped with semantic meanings for their nodes, edges, and even themselves as a whole. For example, the nodes in a molecular graph are atoms and the nodes in a citation graph are papers. They all can be described by text. Hence, we use an LLM to process the textual information, $t_i$, of a node $i$, yielding the node feature vector

$$\mathbf{x}_i = \mathrm{LLM}(t_i). \tag{3}$$

An advantage of obtaining node features in this way is that the pre-trained LLM unifies knowledge across domains and provides the same feature dimension for all nodes of any domain. For non-textual graphs, which are not the focus of this work, one could apply LLMs to generate node descriptions or use projection/padding methods to match the hidden dimension of RWPT. However, since RWPT is pre-trained primarily on text-attributed graphs, such approaches may weaken the alignment induced by the LLM and lead to less competitive results compared to text-rich graphs. A case study is included in Section H.6.

Similarly, for an edge $ij$ of a graph and the virtual token $\mathbf{v}$ of a graph dataset, let their textual description be $t_{ij}$ and $t_v$. Then, we obtain the edge feature and virtual-node feature

$$\mathbf{e}_{ij} = \mathrm{LLM}(t_{ij}), \qquad \mathbf{v} = \mathrm{LLM}(t_v), \tag{4}$$

respectively. The virtual-node feature $\mathbf{v}$ will be used together with node features in the input sequence; the use of the edge feature $\mathbf{e}_{ij}$ will be elaborated later.

**Positional encoding.** We enhance the integration of the graph structure by leveraging positional encodings based on shortest-path (SP) distances. Specifically, for a node $i_s^r$ in the rooted sequence $\mathrm{seq}(i_0)$, its position is defined as the SP distance from the root $i_0$ to $i_s^r$, which is at most $s$. Additionally, for the virtual token $v$ and the root token $i_0$, their position is 0.

In Section 4, we will analyze the representation power of random walks equipped with the positional encoding. In particular, the positional encoding can be used to extract SPs from the walks: if the positions of nodes on a walk segment from $u$ to $v$ are monotonically increasing by 1, then this segment must be a shortest path between $u$ and $v$.

**Incorporating edge features.** Edge features can be used to enhance the encoding of a node sequence. For the $r$th walk $i_0, i_1^r, \ldots, i_\ell^r$, we form an edge-feature sequence

$$\mathbf{E}^r = [\mathbf{e}_{i_0, i_1^r}, \mathbf{e}_{i_1^r, i_2^r}, \ldots, \mathbf{e}_{i_{\ell-1}^r, i_\ell^r}], \tag{5}$$

and we concatenate the $k$ walks and prepend two zero vectors to form the full sequence

$$\mathbf{E} = [\mathbf{0}, \mathbf{0}, \mathbf{E}^1, \mathbf{E}^2, \ldots, \mathbf{E}^k], \tag{6}$$

which has the same length as the Transformer input.

Rather than merely adding $\mathbf{E}$ to the Transformer input, we project $\mathbf{E}$ and add it to the input of each Transformer block, similar to how the edge information is processed in every layer of a GNN (Hu et al., 2020a). Specifically, let the $t$th block of a standard Transformer (Vaswani et al., 2017; Liu et al., 2018) be $\mathbf{H}^{(t+1)} = \mathrm{Block}(\mathbf{H}^{(t)})$. We introduce a block-dependent projector (a linear layer), Proj, and modify the block to be $\mathbf{H}^{(t+1)} = \mathrm{Block}(\mathbf{H}^{(t)} + \mathrm{Proj}^{(t)}(\mathbf{E}))$. The projectors are mainly used to map data from the LLM output dimension to the Transformer embedding dimension, similar to the one for node features. Edge features are incorporated only for graph-level tasks to prevent potential label leakage.

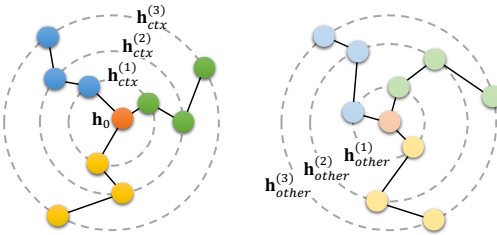

Figure 2: Per-walk attention mask. Each random walk only attends to nodes within its own sequence (diagonal blocks are colored differently). The virtual dataset node ($v$) and the root node (0) are unmasked globally for all random walks. This design reduces the computational complexity by a factor of $k$.

Figure 3: Context learning. The mutual information between the root node embedding ($h_0$) and its aggregated context embedding ($h_{ctx}^{(j)}$) is maximized. Simultaneously, the mutual information with context sampled from other nodes or graphs ($h_{other}^{(j)}$) is minimized. Dashed lines represent various context windows.

### 3.3 PER-WALK ATTENTION MASK

The query-key-value (QKV) attention mechanism typically comes with a mask on the QK product (attention) matrix before softmax. The effect of this mask is to ignore some V items when linearly combining them. For example, the upper triangular part of the attention matrix is masked out in a causal Transformer, because a token will depend on the past but not the future information.

In our case, we use a per-walk attention mask to separate the walks and reduce computation, as illustrated in Figure 2. Each random walk will attend to itself but not each other. The virtual token and the root node will still attend to all the tokens. Clearly, with such a mask, the number of nonzeros is reduced by nearly a factor of $k$ and so is the computational cost. To implement this efficiently, we developed a loop-based kernel where, in each iteration, only one block and the prefix are involved in the computation. This yields a 15% speedup (for ArXiv) compared to applying PyTorch's built-in Flash Attention directly with the per-walk mask.

### 3.4 SELF-SUPERVISED PRE-TRAINING: CONTEXT PREDICTION

Let the Transformer output a sequence of vectors corresponding to the input sequence $s(i)$ in (2):

$$[\mathbf{h}_v, \mathbf{h}_0, \mathbf{h}_1^1, \ldots, \mathbf{h}_\ell^1, \mathbf{h}_1^2, \ldots, \mathbf{h}_\ell^2, \ldots, \mathbf{h}_1^k, \ldots, \mathbf{h}_\ell^k]. \tag{7}$$

The vector $\mathbf{h}_0$ is the representation of the root node $i$.

The pre-training makes use of the output sequence (7) but not task labels. In contrast to the next-token prediction in LLMs, self-supervised learning of GNNs is more often done in a contrastive manner. We follow the infomax principle (Hjelm et al., 2019; Gutmann & Hyvärinen, 2012) and develop a contrastive loss suitable for random walks.

The idea is to define increasingly large context windows for the root node $i$ (see Figure 3):

$$\mathbf{h}_{ctx}^{(j)} = \frac{1}{jk} \sum_{s=1}^{j} \sum_{r=1}^{k} \mathbf{h}_s^r. \tag{8}$$

Here, $\mathbf{h}_{ctx}^{(j)}$ is the representation of the $j$th context window, which includes the nodes up to the $j$th step of all random walks. We maximize the mutual information between the root node $i$ and its context windows, while minimizing that between $i$ and the context windows of other root nodes in the batch. We use an MLP to parameterize the mutual information, leading to the sample loss formula

$$\mathcal{L}_{sample} = -\frac{1}{\ell} \sum_{j=1}^{\ell} \Big( \log \text{MLP}(\mathbf{h}_0 \odot \mathbf{h}_{ctx}^{(j)}) + \sum_{\forall other} \log(1 - \text{MLP}(\mathbf{h}_0 \odot \mathbf{h}_{other}^{(j)})) \Big). \tag{9}$$

This loss encourages the representation of the root node to be close to its neighborhood but different from other neighborhoods.

**Dataset mixture.** Because RWPT is pre-trained with multiple graph datasets, which may vary in size, we introduce a multiplier $\alpha_D$ for each dataset $\mathcal{D}$ of size $n_D$. The batch training consists of multiple passes, each of which iterates over all datasets. For each dataset, a total of $\alpha_D n_D$ nodes are randomly sampled to form batches.

## 3.5 DOWNSTREAM ADAPTATION

With pre-training in place, downstream tasks only require training a lightweight task head while keeping the pre-trained Transformer frozen. For example, for node classification, the task head is an MLP that takes the node representation as input and outputs the class logits; for link prediction, the input to the MLP task head is a concatenation of the two node representations; and for graph classification, the input is the aggregation of node representations (see Section B for details). Besides classification, this downstream adaptation also supports regression tasks (see, e.g., Table 8).

Following the practice of LLMs, other adaptation approaches are exploitable, such as fine-tuning all the parameters of the pre-trained model, fine-tuning only the node and edge feature projectors, or using low-rank adaptors. A comprehensive comparison of these variants is outside the scope of this work; in practice, we find that a lightweight task head suffices to obtain strong results.

## 4 THEORETICAL ANALYSIS

The main idea of this work is to leverage random walks for node representation learning. Combined with SP distance positional encoding, these walks enable the reconstruction and differentiation of local neighborhoods. As a consequence, the pre-trained RWPT matches the expressivity of a certain graph isomorphism test. We sketch the mathematical framework below, with detailed proofs given in Section C. A graph is denoted as $G(V, E)$ with the node set $V$ and the edge set $E$.

**Definition 4.1.** The *Shortest Path Distance Oracle* (*SP oracle*) is a function $\psi : V \times V \to \mathbb{R}$ that takes a pair of nodes as input and returns the shortest path distance between these two nodes.

**Definition 4.2.** Denote by $\mathcal{B}_{u,r} \subset G$ a *ball* centered at node $u$ with radius $r$. Formally, it is a subgraph of $G$ with the node set $V(\mathcal{B}_{u,r}) := \{v \in V \mid \psi(u, v) \leq r\}$ and the edge set $E(\mathcal{B}_{u,r}) := \{e \in E \mid V(e) \subset V(\mathcal{B}_{u,r})\}$, where $\psi(\cdot, \cdot)$ is the SP oracle.

**Definition 4.3** (Grover & Leskovec (2016)). A *Biased Random Walk with parameters $p$ and $q$* is a random walk such that after transitioning from node $u$ to node $v$, the unnormalized probability to return to $u$ is $1/p$, that to jump to a direct neighbor of $u$ is 1, and that to jump to other neighbors of $v$ is $1/q$. The usual (unbiased) random walk is recovered with $p = q = 1$.

The following theorem states that a ball can be reconstructed by a sufficient number of random walks together with the SP distance positional encoding.

**Theorem 4.4.** *Assume that the graph $G$ is undirected and connected, with a bounded degree $d$. Let a ball $\mathcal{B}_{u,r}$ with center $u$ and radius $r$ have $n$ nodes. The ball can be fully reconstructed given the sequence* (1) *together with the SP distance of every node from the root $u = i_0$, by using $k = \Theta(\max(nr, n^2/r^2))$ walks of length $\ell = \Theta(r)$ in expectation.*

Theorem 4.4 implies that users can adjust $\ell$ to control the coverage radius of the local subgraph and then select $k$ accordingly to ensure reliability of structural information. In practice, $k$ is often small, especially when the graph is sparse or incomplete. An ablation study on the choice of $k$ is given in Section 5.4. Here, we focus on the theoretical case.

Because random walks can reconstruct a ball, two balls can be distinguished with a graph kernel.

**Theorem 4.5.** *There exists a positive definite kernel function that distinguishes non-isomorphic balls centered at different nodes of $G$.*

The above result suggests that the multiple random-walk input format preserves reliable structural information. The expressive representation can be translated to the expressivity of the model in distinguishing non-isomorphic graphs. The following conclusion is a straightforward consequence of Theorems 4.4 and 4.5 of Zhang et al. (2023) and it suggests that RWPT matches the expressivity of Graphormer-GD, a GT proposed by Zhang et al. (2023).

**Corollary 4.6.** *Using the walk count $k$ and walk length $\ell$ analyzed in Theorem 4.5 and sufficiently many heads and layers, RWPT is as expressive as the Generalized Distance Weisfeiler–Lehman test.*

Table 1: Performance comparison of (semi-)supervised, self-supervised, and unified model methods across domains and tasks. **Bold** and underline denote the best and second-best results. "OOM" indicates out of memory, and "-" denotes cases where the setting is not applicable (e.g., no prompts available). Baseline results are replicated from Wang et al. (2024).

| | NODE CLASSIFICATION | | | | LINK PREDICTION | | GRAPH CLASSI. | |
| | CORA | PUBMED | ARXIV | WIKICS | WN18RR | FB15K237 | HIV | PCBA |
|---|---|---|---|---|---|---|---|---|
| GCN | 75.65 | 75.61 | 71.40 | 75.28 | 73.79 | 82.22 | 64.84 | 71.32 |
| GIN | 73.59 | 69.51 | 65.05 | 49.77 | 74.02 | 83.21 | 66.86 | 70.12 |
| GAT | 76.24 | 74.86 | 70.87 | 76.78 | 80.16 | 88.93 | 65.54 | 72.69 |
| GPS | 71.51 | 73.60 | OOM | 78.40 | OOM | 93.76 | **75.39** | 79.90 |
| DGI | 72.10 | 73.13 | 69.15 | 75.32 | 75.75 | 81.34 | 59.62 | 63.31 |
| BGRL | 71.20 | 75.29 | 71.19 | 76.53 | 75.44 | 80.66 | 63.95 | 67.09 |
| GRAPHMAE | 73.10 | 74.32 | 70.90 | 77.61 | 78.99 | 85.30 | 61.04 | 63.30 |
| GIANT | 75.13 | 72.31 | 70.10 | 76.56 | 84.36 | 87.45 | 65.44 | 61.49 |
| FUG | **80.63** | **77.34** | OOM | 78.19 | 77.91 | 86.60 | 66.34 | 66.34 |
| GRAPHCLIP | 78.25 | 76.18 | 73.72 | 77.89 | - | - | - | - |
| GFT | 78.62 | 77.19 | 71.93 | 79.39 | 91.91 | 89.72 | 72.67 | 77.90 |
| RWPT | 79.30 | 74.97 | **75.14** | **80.27** | **95.25** | **95.23** | 75.15 | **81.03** |

## 5 EXPERIMENTS

In this section, we present a comprehensive set of experiments to evaluate the effectiveness of RWPT on a variety of graph learning tasks, highlighting transferability in cross-domain/cross-task settings.

### 5.1 EXPERIMENT SETUP

**Datasets.** We use 14 datasets from diverse domains and for varying tasks. They include those supporting node-level tasks (Cora, CiteSeer, PubMed, Arxiv, WikiCS, and Products, where the first four are **citation networks** and the next two are the **Web graph** and the **co-purchase graph**, respectively); those supporting link-level tasks (WN18RR and FB15k237, which are **knowledge graphs**); and those supporting graph-level tasks (HIV, PCBA, ChEMBL, and Tox21, which are **molecules**). We also include Peptides-func and Peptides-struct (also molecules) from the **Long Range** Graph Benchmark (Dwivedi et al., 2022). Altogether, these datasets contain 25M nodes and 31M edges. See Section E for more details.

**Baselines.** We compare RWPT with ten methods in diverse nature, including PRODIGY (Huang et al., 2023), OFA (Liu et al., 2024a), FUG (Zhao et al., 2024b), GraphCLIP (Zhu et al., 2025), and GFT (Wang et al., 2024), **which are pre-trained models with transferability**; GCN (Kipf & Welling, 2017), GIN (Xu et al., 2019), GAT (Veličković et al., 2018), and GPS (Rampášek et al., 2022) **which are task-specific (semi-)supervised models**; and DGI (Veličković et al., 2019), BGRL (Thakoor et al., 2022), GraphMAE (Hou et al., 2022), and GIANT (Chien et al., 2022), **which are self-supervised training methods**. Note that OFA differs in the usual sense in that it does not have a label-free pre-training stage, but we categorize it together with PRODIGY and GFT to distinguish it from the remaining methods that train a different model for each dataset.

**Settings.** Our Transformer backbone follows a standard architecture like GPT-2 (Radford, 2018), with modifications introduced in Section 3 and hyperparameters detailed in Section F. We utilize Llama2-7b (Touvron et al., 2023) for feature extraction; the prompts can be found in Section G. All experiments are conducted with 2x NVIDIA Tesla V100 16GB GPUs, Intel Xeon Platinum 8260 CPUs (32 cores), 50GiB RAM, and 1TB user storage space. Each run is repeated ten times with random seeds.

### 5.2 CROSS-DOMAIN AND CROSS-TASK PERFORMANCE

We compare RWPT with a broad set of methods across domains and tasks, including node-level, link-level, and graph-level predictions. The baselines span (semi-)supervised methods, self-supervised approaches, and models pre-trained across datasets. The first two groups are trained on a

Table 2: Transfer learning performance. † denotes the best performance among all (semi-)supervised methods in Table 1; ‡ denotes the best performance among all self-supervised methods in Table 1; * denotes pre-training RWPT with Arxiv + FB15k237 + ChEMBL.

| | CORA | ARXIV | WIKICS | (AVG.) | WN18RR | FB15K237 | (AVG.) | HIV | PCBA | (AVG.) |
|---|---|---|---|---|---|---|---|---|---|---|
| (SEMI-)SUPERVISED† | 76.24 | 71.40 | 76.78 | 75.01 | 80.16 | 88.93 | 84.55 | 66.86 | 72.69 | 69.78 |
| SELF-SUPERVISED‡ | 75.13 | 71.19 | 77.61 | 74.81 | 84.36 | 87.45 | 85.91 | 65.44 | 67.09 | 66.27 |
| PRE-TRAIN W/ ARXIV | 76.63 | 75.07 | 79.95 | 76.00 | 94.62 | 94.08 | 94.35 | 73.46 | 79.69 | 76.58 |
| PRE-TRAIN W/ FB15K237 | 73.71 | 74.40 | 79.50 | 75.29 | 94.34 | 94.82 | 94.58 | 73.57 | 78.95 | 76.26 |
| PRE-TRAIN W/ CHEMBL | 70.34 | 74.80 | 79.27 | 74.28 | 93.97 | 93.46 | 93.72 | 74.85 | 80.58 | 77.72 |
| PRE-TRAIN W/ THREE* | 78.69 | 75.07 | 80.15 | 76.57 | 94.53 | 95.16 | 94.85 | 74.03 | 81.14 | 77.59 |
| PRE-TRAIN W/ ALL | 79.30 | 75.14 | 80.27 | 77.42 | 95.25 | 95.23 | 95.24 | 75.15 | 81.03 | 78.09 |

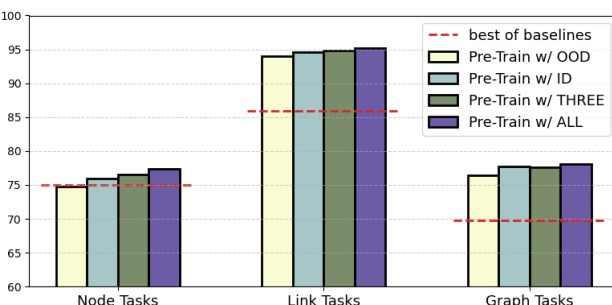

Figure 4: Aggregated transfer learning performance. "Best of baselines" denotes the highest score among (semi-)supervised and self-supervised methods. "OOD" (resp. "ID") indicate that the datasets for pre-training and downstream testing are from the same (resp. different) domain. "Pre-Train w/ THREE" and "Pre-Train w/ ALL" follow the definitions in Table 2.

single dataset and are therefore not directly comparable to RWPT, but they provide useful reference points. Following GFT, we pre-train RWPT on ten datasets (see Section F for batching ratios) and then fine-tune a task head for each downstream task.

Table 1 indicates that pre-trained models achieve better results than task-specific models on seven out of eight tasks. Within the pre-trained model category, RWPT performs the best on six tasks, indicating strong and consistent performance across domains and tasks.

## 5.3 TRANSFERABILITY

While the outperformance of RWPT over individually trained models is not surprising, we investigate in depth its transferability. For this, we consider transfer learning and few-shot learning.

**Transfer learning (dataset- and domain-level transfer).** This learning paradigm tests a pre-trained model on an unseen dataset or domain. To this end, we pre-train RWPT with limited datasets and evaluate it on others. Specifically, we use either Arxiv, FB15k237, ChEMBL, or a combination of them and compare with the early use of ten datasets. The three datasets represent different domains: citation, knowledge graph, and molecules.

The results are summarized in Table 2 and Figure 4. We see that using three datasets to pre-train achieves a performance very close to using ten. This suggests that a small amount of representative datasets are already competitive for pre-training, demonstrating the transferability of RWPT to new datasets. Interestingly, pre-training on only a single dataset also yields strong results, often outperforming models individually trained from scratch.

To highlight domain transfer, Figure 4 compares aggregated in-domain (ID) and out-of-domain (OOD) results. For instance, pre-training on Arxiv and testing on knowledge graphs or molecules constitutes OOD, while testing on citation graphs is ID. The gap between OOD and ID is small relative to the best baselines, indicating that our pre-training method supports meaningful transfer across domains. Random walk patterns play a key role in enabling this cross-domain generalization.

**Few-shot learning (label-level transfer).** In few-shot learning, a support set of $N$ classes with $k$ examples is provided; the goal is to evaluate the ability of learning from limited data.

Table 6 in Appendix reports results from $N$-way $k$-shot experiments across several datasets. We compare our model, RWPT, with (i) methods trained separately per dataset and (ii) pre-trained

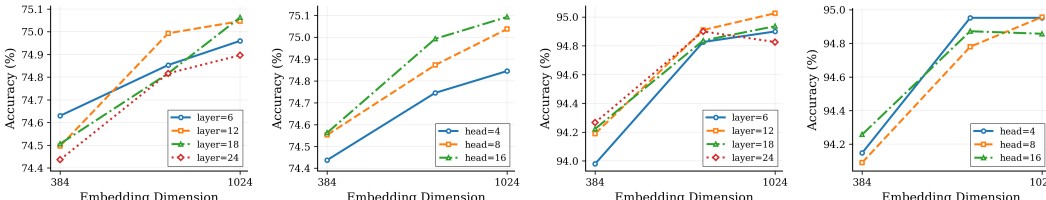

Figure 5: Ablation study on Transformer configuration. Each plot shows performance as embedding dimension increases. The left two plots report results on ArXiv node classification, and the right two on FB15K237 link classification. Each curve corresponds to a different number of layers or attention heads.

models fine-tuned across domains. Unsurprisingly, pre-trained models generally outperform task-specific models due to data scarcity. RWPT achieves the strongest results in most cases.

## 5.4 ABLATION STUDY

**Comparison between random-walk sampling and neighborhood sampling.** We motivated the use of random walks for representation learning in Section 1 with a few reasons. One of the reasons is that random walks can better cope with long-range interactions if they are important for some downstream tasks or datasets. Compared with neighborhood sampling (Hamilton et al., 2017), multiple random walks equally retain sufficient near-hop neighbors while being able to extend very far.

From Table 8 in Section H.1, we observe that no single setting consistently outperforms others. However, we note that random walks tend to outperform neighborhood sampling on the two datasets from the Long Range Graph Benchmark (Peptides-func, Peptides-struct). The optimal walk length varies by dataset, but random walks offer a flexible way to capture broader receptive fields.

**Comparison of LLM text encoders and Transformer configurations.** We conduct an ablation study with different LLMs for text feature encoding, varying both model sizes and embedding dimensions. Results are reported in Table 9 in Section H.2. Overall, stronger LLMs with higher-dimensional embeddings generally lead to better downstream performance and certain models show task-specific advantages such as E5 over molecular tasks.

Regarding the configuration of the Transformer model, in Figure 5, we observe that increasing the embedding dimension generally improves performance, especially on larger training datasets. In contrast, adding more layers or heads does not show a clear benefit. Based on this, we recommend scaling the embedding dimension with the dataset size, while keeping the number of layers and attention heads moderate.

**Comparison of training losses.** We pre-train our RWPT using an InfoNCE-style contrastive loss (9), which compares each node's random-walk context against those of other nodes. We compare this objective with prior approaches, including DGI (Veličković et al., 2019), GraphPrompt (Liu et al., 2023b), and MaskGAE (Li et al., 2023), as well as a masked-token reconstruction variant inspired by language model objectives. Results in Tables 10 (Section H.3) and 11 (Section H.4) show that our contrastive loss achieves better overall performance on several datasets. Adding mask reconstruction slightly change rankings but does not yield consistent gains. Overall, the results suggest our loss is a competitive choice for pre-training text-attributed graphs.

## 6 CONCLUSIONS

We introduced a Transformer-based approach for graph learning that leverages random walks as input sequences and a context prediction objective for self-supervised learning. By combining data from various domains, the proposed model, RWPT, can be adapted with simple fine-tuning to different downstream tasks and graphs. Our experiments show that RWPT achieves performance comparable to graph models trained specifically on individual datasets and those trained on multiple datasets in a supervised manner.

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

# Appendix

## A    MORE RELATED WORK

**Graph sequence models.** Behrouz et al. (2024) proposed the concept of Graph Sequence Models (GSMs), which translate graphs into sequences through tokenization, local, and global encodings. Many works such as Graph-Mamba (Behrouz & Hashemi, 2024) and GraphGPS (Rampášek et al., 2022) fit within this framework. Our RWPT model aligns with GSMs by combining (1) random-walk tokenization, (2) local encodings derived from LLM-processed text features, and (3) global encoding via a transformer. Moreover, unlike most task-specific models mentioned in Behrouz et al. (2024), RWPT is designed for pre-training across diverse text-attributed graphs to enable flexible fine-tuning on downstream tasks.

**Graph transformers.** In addition to the GTs predominantly appearing in the graph literature (Section 1), the NLP community also explored Transformer-based architectures for structured data. G2GTr (Mohammadshahi & Henderson, 2019) and RNGTr (Mohammadshahi & Henderson, 2021) adapt Transformers to handle graph-structured input and output in dependency parsing tasks. Although not originally designed for general graphs, these models incorporate structure-aware mechanisms that capture local connectivity within sequential input, typically consisting of redundant neighborhood nodes. Our method shares a similar intuition with these methods.

**Pre-training GNNs.** Pre-training is essential to LLMs and to a degree, GNNs. An early GNN pre-training study (Hu et al., 2020b) develops a multitude of loss terms that, when invoked together, incur positive transfer and significantly improve task performance, while naive pre-training methods incur negative transfer. However, some of the loss terms exploit task labels (e.g., graph property prediction). Nevertheless, another loss term—context prediction—is particularly useful and our method is inspired by this idea. Additionally, a proliferation of self-supervised learning methods were developed for GNNs in the past (Veličković et al., 2019; Zhu et al., 2020; You et al., 2020; Zhu et al., 2021; Thakoor et al., 2022; Hou et al., 2022; Chien et al., 2022). They could be used for pre-training and we compare some of them in the experiments.

**Unified GNN models.** LLMs bring in new concepts that motivate the modern development of GNNs. Prompting is one example. GPPT (Sun et al., 2022) pre-trains a usual GNN and introduces a prompting function to adapt the GNN to produce (node, label) embedding pairs for node classification. GraphPrompt (Liu et al., 2023b) incorporates learnable prompts and a task-unified objective to bridge the gap between pre-training and fine-tuning in GNNs, enabling the model to extract task-relevant information from a pre-trained model. PRODIGY (Huang et al., 2023) focuses on the few-shot learning setup, which builds a graph that connects a few prompt examples (the shots), the query, and the task labels for classifying the query. FUG (Zhao et al., 2024b) learns a feature-unified encoder through a PCA-like basis transformation and contrastive learning framework, enabling alignment of diverse node feature dimensions. ULTRA (Galkin et al., 2023), designed for knowledge graphs, leverages relative entity and relation embeddings for inductive and transductive reasoning. In contrast, our approach does not pre-train on relations in link-prediction datasets, because relations are inherently tied to the link-prediction downstream task. Instead, we concatenate head and tail node embeddings to predict the relation—a simpler yet effective strategy.

Several recent works aim to unify the feature space across graph datasets. All-in-One (Sun et al., 2023) trains prompt tokens derived from subgraphs and inserts them into node representations, enabling a pre-trained GNN to adapt to multiple tasks. Zhao et al. (2024a) propose a prompt-based method for cross-domain classification, where features are aligned through singular value decompositions. OpenGraph (Xia et al., 2024) introduces a topology-aware graph tokenizer that converts graphs into sequences of unified embeddings, which are then processed by a scalable graph transformer. In contrast, our model aligns datasets using a pre-trained LLM without requiring a dataset-specific tokenizer—though such tokenizers remain compatible for task-specific adaptations. For text-attributed graphs, OFA (Liu et al., 2024a), GFT (Wang et al., 2024), UniGraph (He & Hooi, 2024), and GraphCLIP (Zhu et al., 2025) leverage pre-trained LLMs to unify textual features. We adopt a similar strategy to bridge cross-domain graph representations. Additionally, inspired by GCC (Qiu et al., 2020), we incorporate contrastive learning on subgraph patterns to enhance cross-task discrimination.

**LLM-based methods.** Several LLM-centric approaches have been proposed for graph tasks, where graph data is converted into text-compatible formats that LLMs can process. GraphPrompter (Liu et al., 2024b) and GNP (Tian et al., 2024) integrate GNN embeddings as soft prompts into text tokens, enabling LLMs to handle graph-structured inputs. Other studies (Wang et al., 2023; Fatemi et al., 2024) show that LLMs can capture structural properties through natural language descriptions of nodes and their connections. GPT4Graph (Guo et al., 2023) assesses the ability of LLMs to understand and reason over structured graph data. GraphLLM (Chai et al., 2023) augments LLMs with a graph transformer to improve graph reasoning. GraphText (Zhao et al., 2023) transforms graph data into natural language sequences using graph-syntax trees, allowing in-context reasoning without fine-tuning. GraphGPT (Tang et al., 2024) aligns graph and textual representations via dual-stage instruction tuning, improving both supervised and zero-shot performance. LLaGA (Chen et al., 2024b) translates graphs into LLM-readable formats to support prompt-based task execution. In contrast, our model employs a lightweight task-specific prediction head without fine-tuning the LLM, resulting in a compact encoder comparable in size to GPT2-124M. Similarly, WalkLM (Tan et al., 2023) uses textual random walks to construct LLM input and contrastive pre-training with task-specific heads; however, it does not support joint training across domains. For graph-to-sequence methods, GraphGPT (Zhao et al.) uses reversible Eulerian-path, which provides theoretical guarantees but incurs high pre-processing cost. In contrast, our RWPT adopts a more practical random-walk approach. Unlike GraphGPT's reliance on an explicit embedding table, we leverage pre-trained LLM features, which offer broader cross-domain transfer on text-attributed graphs. Readers can refer to Yuan et al. (2025); Liu et al. (2023a) for a more detailed survey.

## B  DOWNSTREAM ADAPTATION

The pre-trained model needs to be adapted to perform a downstream task. In this work, we take a simple form of adaptation by using task heads. Specifically, a task head is a neural network that takes the pre-trained model output as input and performs the downstream task. Because we perform evaluation in multiple scenarios (cross-domain cross-task learning, transfer learning, and few-shot learning), the task heads include some of the following components: encoder, decoder, and scorer.

- The **encoder** maps a generic node representation (the pre-trained model output $h_0$ in (7)) to a task-specific node representation. The encoder is an MLP, where each layer consists of four components: linear transformation, batch normalization, ReLU activation, and dropout. For node-level tasks, the input consists of node representations. For link-level tasks, the input is the concatenation of the representations of the two nodes forming the edge. For graph-level tasks, the input graph representations is computed by aggregating node representations across the entire graph. Hyperparameters are detailed in Section F.

- The **decoder** maps task-specific node representations to task targets. Note that concatenation for link tasks and pooling for graph tasks are performed within the encoder. Therefore, the decoder is always a single linear layer, where the inputs are the outputs of the encoder. The output dimension of the decoder corresponds to the number of classes for classification and a single digit for regression.

- The **scorer** is an unparameterized module responsible to compare two representations. In few-shot learning, this scorer will compare the query-node representation and the class representation. We use the cosine similarity for comparison and apply a softmax on the vector of similarities for label prediction. The class representation is computed as the average of the examples in the support set.

For the cross-domain cross-task learning and transfer learning, the task head is the encoder–decoder pair. Each task head is trained by using the training set of the corresponding task with labels.

For few-shot learning, we first train the encoder–decoder pair by using very few examples. Then, we use the trained encoder and pair it with the scorer for prediction. It was suggested that this approach is better than directly training the encoder–scorer pair (Huang et al., 2023; Wang et al., 2024) and we confirm so in preliminary tests.

## C    Proofs for Section 4

### C.1    Proof of Theorem 4.4

**Lemma C.1.** *The expected walk length for reaching the farthest node in $B_{u,r}$ is $\mathbb{E}[\ell] = O(r)$ when*

$$\frac{1}{q} \geq \frac{C+1}{C-1} \cdot \frac{1}{p} + \frac{1}{C-1}d, \tag{10}$$

*for some constant $C \geq 2$ and $C \ll r$.*

*Proof.* Let us consider three consecutive nodes in a walk $(v_0, v_1, v_2)$ with $\psi(u, v_0) = i - 1$ and $\psi(u, v_1) = i$. For the transition from $v_1$ to $v_2$, we assume the probability of moving back, i.e., $v_2 = v_0$, is $p_b$; the probability of moving outward to a node with $\psi(u, v_2) = i + 1$ is $p_f$; and the probability of moving in the same depth is $1 - p_b - p_f$.

Following Lemma 4.7 in Blum et al. (2015), we denote $h_{i,i+1}$ as the hitting time from a node of distance $i$ to a node of distance $i + 1$. Then,

$$h_{i,i+1} = p_f \cdot 1 + p_b(1 + h_{i-1,i} + h_{i,i+1}) + (1 - p_f - p_b)(1 + h_{i,i+1}). \tag{11}$$

Since the initial state $h_{0,1} = 1$, solving above equation yields the following recursion:

$$h_{i,i+1} = \frac{1 + p_b h_{i-1,i}}{p_f} = \left(\frac{p_b}{p_f}\right)^i + \frac{1}{p_b} \sum_{t=1}^{i} \left(\frac{p_b}{p_f}\right)^t. \tag{12}$$

Consider the worst case of the biased random walk under the degree bound assumption, where we have $p_f = p/(p+q+pqd)$ and $p_b = q/(p+q+pqd)$. Assumption (10) implies $p_f > p_d$; therefore, we can define $\alpha := p_b/p_f = q/b < 1$. Then,

$$h_{0,r} = \sum_{i=0}^{r-1} h_{i,i+1} = (1 - \frac{1}{p_f - p_b})\frac{1 - \alpha^r}{1 - \alpha} + \frac{r}{p_f - p_b}. \tag{13}$$

Since $1 > p_f > p_b > 0$, the first term is negative, we only need to consider the second term. From Eq. (10), we have $p_f - p_b \geq 1/C$, which leads to:

$$h_{0,r} \leq \frac{r}{p_f - p_b} \leq C \cdot r. \tag{14}$$

$\square$

The above lemma indicates that the expected number of random walk steps to traverse from the root node to a node with distance $r$ is $O(r)$, with proper choices of $p$ and $q$.

**Corollary C.2.** *When $p = q$, the expected walk length for reaching the farthest node in $\mathcal{B}_{u,r}$ is $\mathbb{E}[\ell] = \Theta(r^2)$.*

*Proof.* Starting from Eq. (12) with $p_b/p_f = 1$, we obtain

$$h_{i,i+1} = 1 + \frac{i}{p_b} \quad \rightarrow \quad h_{0,r} = \sum_{i=0}^{r-1} h_{i,i+1} = r + \frac{1}{p_b}\frac{r(r-1)}{2} = \Theta(r^2). \tag{15}$$

$\square$

The above corollary indicates that for an unbiased random walk, the expected walk length $\ell$ is of the same order as Lemma 4.7 of Blum et al. (2015), up to a constant factor.

**Corollary C.3.** *When $p < q$, the expected walk length for reaching the farthest node in $\mathcal{B}_{u,r}$ can be exponential in $r$.*

The above corollary indicates that in rare cases, where a random walk favors moving back over moving outward, the expected walk length can grow extremely fast.

Here are the interpretations of the results so far. (1) When the random walk has equal probability of moving inward or outward, the expected hitting time has the same order as that of an unbiased random walk. (2) When the random walk favors returning rather than progressing, the hitting time can grow exponentially. (3) In most practical cases, where the walk prioritizes depth over breadth and backward steps, the hitting time is linear and efficient to cover enough range.

Empirically, the required walk length $\ell$ indeed grows linearly. For example, the Cora dataset has an average node degree $d \approx 4$. We set $p = 1$ and $q = 0.1$ to favor moving outward over backward. Assume that the desired $r = 7$. Lemma C.1 suggests a walk length of $\ell = 12$; meanwhile, we use $\ell = 8$ and observe that nodes at distance 7 are frequently reached.

Before reconstructing $\mathcal{B}_{u,r}$, we show that the input sequence (1) can provide enough distinct SP oracles $\psi(v_1, v_2)$ within $\mathcal{B}_{u,r}$.

**Lemma C.4.** *Assume that $\mathcal{B}_{u,r}$ contains $n$ nodes. Then, the expected number of random walks with $\ell = \Theta(r)$ to cover all nodes in $\mathcal{B}_{u,r}$ is*

$$\mathbb{E}[k] = O(nr). \tag{16}$$

*Proof.* According to Lemma C.1, a random walk $w$ of length $\ell$ is expected to yield at least $r$ distinct SP oracles $\psi(u, v)$ where $v \in V(w)$ and $u$ is the root node. This provides at least as much information as a depth-first search (DFS) walk of length $r$, ignoring nodes with the same depth.

Let us consider a DFS tree with root node $u$ and depth $r$. Assume it has $m_r$ nodes. Clearly, $m_r \leq \min(n, d^r)$. Now, we compute the expected number of random DFS walks of length $r$ required to cover all leaves. This is the *coupon collector problem*; hence, $k = \Theta(m_r \ln m_r)$. Similarly, to cover all nodes we need $k = \Theta(m \ln m)$ where $m = \max_i(m_i)$. Therefore, $k = O(mr) = O(nr)$. $\square$

**Lemma C.5.** *With the same assumption in Lemma C.4, given $\ell = \Theta(r)$ and $k = \Theta(n^2/r^2)$, the expected number of distinct SP oracles provided by one input sequence (1) is $\Theta(n^2)$.*

*Proof.* We define the *pseudo SP oracle* given by a random walk $w$ as:

$$\widetilde{\psi}_w(v_1, v_2) := |w(v_1) - w(v_2)|, \tag{17}$$

where $w(v)$ returns the index of node $v$ in walk $w$. Then, by denoting $W = \{w_1, \ldots, w_k\}$, we further define:

$$\widetilde{\psi}(v_1, v_2) := \min_{w \in W} \widetilde{\psi}_w(v_1, v_2). \tag{18}$$

We need to justify the cases where the pseudo SP oracle is accurate and yet reliable. For example, $\widetilde{\psi}(v_1, v_2) = \psi(v_1, v_2)$ if the shortest path between $v_1$ and $v_2$ is traversed by a random walk $w \in W$. Moreover, if $\widetilde{\psi}(v_1, v_2) = 1$, i.e. $v_1, v_2$ are consecutive nodes in a walk, it is also accurate. The smaller $\widetilde{\psi}(v_1, v_2)$ is, the higher the probability that the pseudo SP oracle is accurate.

In general, if the nodes in a subsequence $\underline{w} \subset w$ exhibit strictly increasing depth, i.e.,

$$\psi(u, v_j) = \psi(u, v_{j-1}) + 1, \quad \forall [v_{j-1}, v_j] \subset \underline{w}, \tag{19}$$

then the pseudo SP oracle yields accurate estimates for any pair of nodes within the subsequence $\underline{w}$. Here, $[v_{j-1}, v_j]$ denotes any two consecutive nodes in $\underline{w}$. Consequently, the number of reliable SP oracles is $|\underline{w}|(|\underline{w}| - 1)/2$.

We consider a path of $r + 1$ nodes with depth from $0$ to $r$. We assume the probability of moving forward is $p_f$, which is equivalent to adding a break point at each internal point with probability $1 - p_f$. The expected number of strictly increasing subsequences is: $\mathbb{E}[n_s] = 1 + (r - 1)(1 - p_f)$. Therefore, the expected number of nodes in a subsequence is $\mathbb{E}[\ell_s] = 1 + r/\mathbb{E}[n_s]$. Then, the expected number of SP oracles is:

$$\mathbb{E}[n_{SP}] = \mathbb{E}[n_s]\frac{\mathbb{E}[\ell_s](\mathbb{E}[\ell_s] - 1)}{2} = \frac{r^2}{2 + 2(r - 1)(1 - p_f)} + \frac{r}{2}. \tag{20}$$

Let $p_f = 1 - 1/r$ by tuning the biased random walk. Then we have:

$$\mathbb{E}[n_{SP}] \approx \frac{1}{4}(r+1)^2 = \Theta(r^2), \tag{21}$$

which leads to the conclusion given the assumptions. $\qquad\square$

In practice, $k$ can often be much smaller, particularly when the graph is sparse or far from complete. For instance, in a 2D grid where $n = \Theta(r^2)$, we obtain $k = \Theta(r^2) = \Theta(n)$. The sparser the graph, the smaller $k$ needed. Therefore, as suggested by Lemmas C.4 and C.5, the selection of $k$ should balance two factors: ensuring sufficient coverage of the graph and enough number of accurate SP oracle estimations.

We are now ready to show the main theorem as a consequence of the following propositions.

**Proposition C.6** ((Reyzin & Srivastava, 2007, Table 1)). *The reconstruction of an arbitrary graph requires $\Theta(n^2)$ SP oracles. It can be reduced to $\Theta(dn \log n)$ for trees of bounded degree $d$.*

**Proposition C.7** ((Mathieu & Zhou, 2013, Theorem 1)). *With a constant degree bound $d$, reconstruction complexity can be reduced to $\Theta(n^{3/2})$ with a randomized method.*

**Theorem C.8** (Theorem 4.4). *The ball $\mathcal{B}_{u,r}$, can be reconstructed via the input sequence* (1) *by using $k = \Theta(\max(nr, n^2/r^2))$ walks of length $\ell = \Theta(r)$ in expectation.*

*Proof.* Lemmas C.1, C.4 and C.5 guarantee that with appropriate $k$ and $\ell$, the input sequence (1) can fully cover all nodes in the ball and provide $\Theta(n^2)$ SP oracles. Then, following Propositions C.6 and C.7, we can reconstruct $\mathcal{B}_{u,r}$, which concludes the proof. $\qquad\square$

C.2 PROOF OF THEOREM 4.5

We now show the expressivity of the input sequence (1). We abbreviate a ball as $B_i := B_{u_i,r}$ since the radius does not matter for graph isomorphism. The proof is based on the following proposition.

**Proposition C.9** ((Borgwardt & Kriegel, 2005, Section 4)). *There exists a graph kernel based on SP oracles that is computable in polynomial time, expressive up to isomorphism, and positive definite.*

**Theorem C.10** (Theorem 4.5). *There exists a positive definite kernel function $\Phi$ that can distinguish two balls $\mathcal{B}_1$ and $\mathcal{B}_2$ with $u_1 \neq u_2$, up to isomorphism.*

*Proof.* According to Theorem 4.4, the input sequence (1) can recover SP oracles $\psi$ with appropriate $k$ and $\ell$. Specifically, we can build a subgraph kernel:

$$\Phi(\mathcal{B}_1, \mathcal{B}_2) = \sum_{s_1 \in SP(\mathcal{B}_1)} \sum_{s_2 \in SP(\mathcal{B}_2)} \phi(s_1, s_2) \tag{22}$$

where $SP(\mathcal{B})$ denotes the set of triplets $(v_1, v_2, \psi(v_1, v_2))$ for all $v_1, v_2 \in \mathcal{B}$, and $\phi$ checks if $s_1 = s_2$. The function $\Phi$ is equivalent to the kernel in Proposition C.9 and is therefore positive definite and expressive up to graph isomorphism. $\qquad\square$

**Corollary C.11** (Corollary 4.6). *With appropriate walk count $k$, walk length $\ell$, and sufficiently many heads and layers, RWPT is as expressive as the Generalized Distance Weisfeiler–Lehman (GD-WL) test.*

*Proof.* With suitable choices of $k$ and $\ell$, Theorem 4.5 ensures that our sequential representation, combined with SP distance positional encoding, preserves distance information for injecting into the multi-head attention mechanism. Consequently, RWPT falls under the scope of Theorems 4.4 and 4.5 in Zhang et al. (2023), implying that, with sufficiently many heads and layers, RWPT is as expressive as the GD-WL test and is upper bounded by the 2-FWL test (Cai et al., 1992). $\qquad\square$

# D COST ANALYSIS

A common conception is that Transformers are expensive to train and use. Here, we conduct an analysis to compare the model size and the forward time of RWPT versus GNNs. Surprisingly, under reasonable choices of the hyperparameters, RWPT can be more economical than GNNs.

The main hyperparameters of a Transformer are the context length $L$ and the embedding dimension $d$. Each Transformer block contains four $d \times d$ and two $d \times (4d)$ weight matrices; hence, the per-block model size is dominated by $12d^2$. The main computations are matrix-matrix multiplications of size $(s_m, s_k, s_n)$ per sequence: three $(L, d, d)$, two $(L, d, L)$, and two $(L, d, 4d)$. Thus, the per-block time cost is dominated by $11Ld^2 + 2L^2d$. Overall, for a Transformer with $T$ blocks and a batch size $B$, the approximate model size and forward time are summarized in Table 3.

The main hyperparameters of a GNN with neighborhood sampling are depth $\ell'$ and fanouts $k_1, \ldots, k_{\ell'}$. Reusing the batch size $B$, assume that on average the number of sampled nodes for each aggregation is $s_0(= B), s_1, \ldots, s_{\ell'}$. Further, assume that all the weight matrices are $d' \times d'$. Then, the model size of a GNN is approximately $(d')^2 \ell'$. In the $(\ell' - i)$th layer, there is a matrix-matrix multiplication of size $(s_i, d', d')$ and an aggregation of $k_i$ neighbors for each of the $s_{i-1}$ nodes. Thus, the per-layer time cost is $s_i(d')^2 + s_{i-1}k_i d'$. If we set each $k_i = k'$ and consider the worst case $s_i = s_{i-1}k'$, a simplified upper bound of the overall forward time is given in Table 3.

The size and forward time of the two models are not directly comparable, but we can consider a few cases. If they have the same depth ($T = \ell'$), the same embedding dimension ($d = d'$), and if we allow $L = d$ in the Transformer, then the model size of RWPT is 12 times of GNN, but RWPT is faster than GNN as long as $(k')^T > 13dT$. If instead, we increase the embedding dimension of the GNN to $d' = d\sqrt{12}$, the two models have the same size and RWPT is faster than GNN as long as $(k')^T > \frac{13}{12}dT$.

We report a case study of time cost in each phase. In pre-training, the default setting uses 131M parameters, batch size 200, and sequence length 32 (6.4K nodes per batch). Training typically converges after 300–400 steps, with each step taking 300ms on a single V100 (16GB).

Fine-tuning involves (i) a one-pass RWPT inference on target nodes and (ii) training an MLP prediction head. On Pubmed: random walk sampling requires 36ms, RWPT inference 19s, and prediction head training (2-layer MLP) 40ms per epoch. The overall runtime of RWPT remains within the range of graph transformer and pre-training model baselines. Inference on target nodes can be parallelized or streamed to improve throughput. Notably, random walk generation is negligible relative to transformer training cost.

Table 3: Cost comparison. $B$: batch size; $T$: number of Transformer blocks; $L$: context length; $d$: Transformer embedding dimension; $\ell'$: number of GNN layers; $d'$: GNN embedding dimension; $k'$: fanout.

|  | MODEL SIZE | FORWARD TIME |
|---|---|---|
| RWPT | $12d^2T$ | $(11Ld^2 + 2L^2d)TB$ |
| GNN | $(d')^2\ell'$ | $\frac{k'}{k'-1}(d')^2(k')^{\ell'}B$ |

# E DATASETS

We use a total of 14 datasets for experimentation. Their statistics is summarized in Table 4.

**Cora:** Cora (McCallum et al., 2000) is a citation network comprising 2,708 machine learning publications, each being a node. These nodes are interconnected by 5,429 edges, indicating the citation relationship between papers. The publications come from seven subfields of machine learning. The paper titles and abstracts are available at `https://people.cs.umass.edu/~mccallum/data.html`.

Table 4: Dataset statistics.

| DATASET | DOMAIN | TASK | #GRAPHS | AVG. #NODES | AVG. #EDGES | #TASK | #CLASS |
|---|---|---|---|---|---|---|---|
| CORA | CITATION | NODE | 1 | 2,708 | 5,429 | 1 | 7 |
| CITESEER | CITATION | NODE | 1 | 3,312 | 4,598 | 1 | 6 |
| PUBMED | CITATION | NODE | 1 | 19,717 | 44,338 | 1 | 3 |
| ARXIV | CITATION | NODE | 1 | 169,343 | 1,166,243 | 1 | 40 |
| WIKICS | WEB LINK | NODE | 1 | 11,701 | 216,123 | 1 | 10 |
| PRODUCTS | CO-PURCHASE | NODE | 1 | 54,025 | 144,638 | 1 | 44 |
| ELLIPTIC-BITCOIN | FINANCE | NODE | 1 | 203,769 | 234,355 | 1 | 3 |
| WN18RR | KNOWLEDGE | LINK | 1 | 40,943 | 93,003 | 1 | 11 |
| FB15K237 | KNOWLEDGE | LINK | 1 | 14,541 | 310,116 | 1 | 237 |
| HIV | MOLECULE | GRAPH | 41,127 | 25.45 | 27.47 | 1 | 2 |
| PCBA | MOLECULE | GRAPH | 437,929 | 25.97 | 28.10 | 128 | 2 |
| CHEMBL | MOLECULE | GRAPH | 365,065 | 25.84 | 27.96 | 1,048 | 2 |
| TOX21 | MOLECULE | GRAPH | 7,831 | 18.61 | 19.34 | 12 | 2 |
| PEPTIDES-FUNC | MOLECULE | GRAPH | 15,535 | 150.94 | 307.30 | 10 | 2 |
| PEPTIDES-STRUCT | MOLECULE | GRAPH | 15,535 | 150.94 | 307.30 | 11 | REGRES. |

**citeseer:** citeseer (Giles et al., 1998) is also a citation network, comprising 3,312 scientific publications across six disciplines. Each paper's abstract can be obtained from `https://people.cs.ksu.edu/~ccaragea/russir14/lectures/citeseer.txt`.

**PubMed:** PubMed (Yang et al., 2016) consists of 19,717 papers from the PubMed database, focusing on diabetes research. The text data can be obtained from the TAPE repository (He et al., 2023b).

**Arxiv:** The Arxiv (short for ogbn-arxiv) dataset is a citation network of Computer Science papers submitted to arXiv. It is part of the OGB collection (Hu et al., 2020a) and the paper titles and abstracts can be obtained from the OGB GitHub repository.

**Products:** The Products (short for ogbn-products) dataset is also part of the OGB collection (Hu et al., 2020a). It is an undirected and unweighted graph representing the Amazon product co-purchasing network, containing approximately 2.4 million products and 61.9 million edges. For our experiments, we use the text data from the TAPE repository (He et al., 2023b), which covers only a subgraph of the original ogbn-products.

**WikiCS:** WikiCS (Mernyei & Cangea, 2020) is a graph constructed from Wikipedia, comprising 11,701 articles as nodes and 216,123 hyperlinks as edges. The node features are derived from the articles' text and it is labeled according to the article's subject category. The text data can be obtained from the original paper.

**WN18RR:** WN18RR (Dettmers et al., 2018) is a knowledge graph, comprising 93,003 triples connecting 40,943 entities through 11 distinct relations, sourced from WordNet. The text data can be obtained from `https://github.com/villmow/datasets_knowledge_embedding/tree/master`.

**FB15k237:** FB15k237 (Bordes et al., 2013) is also a knowledge graph, containing 310,116 triples involving 14,541 entities and 237 relation types, derived from Freebase. The text data can be obtained from `https://github.com/villmow/datasets_knowledge_embedding/tree/master`.

**Molecule datasets:** The molecule datasets include HIV, PCBA, and Tox21, which are from MoleculeNet (Wu et al., 2018); ChEMBL (Gaulton et al., 2012); and Peptides-func and Peptides-struct, which are from the Long Range Graph Benchmark (Dwivedi et al., 2022). Each molecule is a graph, where nodes are atoms and edges are chemical bonds. All datasets support multiple tasks, such as predicting biological assays or activities.

**Elliptic Bitcoin:** The Elliptic Bitcoin dataset (Weber et al., 2019) is an example of non-textual graph. The dataset contains a graph with 204k nodes and the task is node classification. Each node is a Bitcoin transaction, represented by 166 preprocessed numerical features whose meanings were deliberately shielded when the dataset was released. Hence, we do not have semantic information to construct LLM features. Each node is labeled as "licit", "illicit" or "unknown".

Table 5: Hyperparameters for downstream tasks. All$^*$ indicates all data in the current split is loaded.

| DATASET | #LAYERS | DIM. | DROPOUT | BATCH SIZE | LOSS | METRIC |
|---|---|---|---|---|---|---|
| CORA | 2 | 256 | 0.5 | ALL$^*$ | NLL | ACC. |
| CITESEER | 2 | 256 | 0.5 | ALL$^*$ | NLL | ACC. |
| PUBMED | 4 | 768 | 0.5 | 2,048 | NLL | ACC. |
| ARXIV | 4 | 768 | 0.5 | 2,048 | NLL | ACC. |
| WIKICS | 2 | 768 | 0.5 | 2,048 | NLL | ACC. |
| PRODUCTS | | | NOT USED IN DOWNSTREAM | | | |
| ELLIPTIC-BITCOIN | 2 | 768 | 0.5 | 2,048 | CE | F1 |
| WN18RR | 3 | 256 | 0.15 | 2,048 | NLL | ACC. |
| FB15K237 | 3 | 256 | 0.5 | 2,048 | NLL | ACC. |
| HIV | 3 | 768 | 0.5 | 2,048 | BCE | AUC |
| PCBA | 3 | 768 | 0.15 | 2,048 | BCE | AUC |
| CHEMBL | | | NOT USED IN DOWNSTREAM | | | |
| TOX21 | 3 | 768 | 0.5 | 2,048 | BCE | AUC |
| PEPTIDES-FUNC | 4 | 768 | 0.15 | 1,024 | BCE | AUC |
| PEPTIDES-STRUCT | 4 | 768 | 0.15 | 1,024 | MSE | MAE |

# F  EXPERIMENT DETAILS AND HYPERPARAMETERS

**Random walks.** We follow Node2Vec (Grover & Leskovec, 2016) and sample biased random walks, which offer greater flexibility in capturing neighborhood information. We set the random walk parameters to $p = 1.0$ and $q = 0.1$, favoring depth over breadth. Without specification otherwise, for each node, we sample $k = 8$ independent walks, each of which has a length $\ell = 4$.

**Transformer.** The Transformer architecture is standard. We use 12 blocks of multi-head attention plus a subsequent 2-layer MLP, with 12 heads per block. The hidden dimension is 768. The projectors for node/edge/dataset features are all linear layers.

**Pre-training.** We employ AdamW (Loshchilov, 2017) as the optimizer with a learning rate `5e-5` and weight decay `1e-2`. We also use a linear learning-rate scheduler with 100 warm-up steps and a scaling factor of 0.1 subsequently. Additionally, we apply gradient clipping with a maximum norm of 1.0 and gradient accumulation of 8 steps.

**Dataset mixture.** We use a maximum of 10 datasets to pre-train RWPT. The multiplier $\alpha_D$ for each dataset $D$ is: PubMed (3.0), Products (0.5), WikiCS (2.0), Arxiv (0.7), WN18RR (0.8), FB15k237 (0.1), PCBA (0.2), ChEMBL (0.1), HIV (1.0), and Tox21 (2.0).

**(Downstream) Cross-domain cross-task learning.** The task heads are MLPs detailed in Section B, with hyperparameteres listed in Table 5. The optimizer is AdamW with a learning rate `1e-3` and all other hyperparameters follow PyTorch's default values.

**(Downstream) Transfer learning.** The hyperparameters are identical to those used in cross-domain cross-task learning. Note that to demonstrate transferability, we vary the pre-training datasets. Hence, the multipliers are set to zero except for those participating in the pre-training. This ensures that no information from non-target datasets is accessed during pre-training.

**(Downstream) Few-shot leanring.** It consists of two stages: encoder–decoder training and encoder–scorer inference (see Section B for more details). The encoder–decoder training phase is similar to cross-domain cross-task learning, with the exception that the number of training examples is very limited. Specifically, for Cora and WN18RR: 1 example per class; Arxiv: 5 examples per class; and FB15K237: 30 examples per class.

**Evaluation metric.** The molecule datasets except Peptides-struct use AUC while Peptides-struct uses MAE; and other datasets use accuracy. See Table 5.

Table 6: Few-shot learning performance. Results of BGRL, GraphMAE, GIANT, PRODIGY, OFA, and GFT are replicated from Wang et al. (2024).

| | ARXIV 5-WAY | | | ARXIV 40-WAY | | | FB15K237 10-WAY | | | FB15K237 40-WAY | | |
|---|---|---|---|---|---|---|---|---|---|---|---|---|
| | 1-SHOT | 3-SHOT | 5-SHOT | 1-SHOT | 3-SHOT | 5-SHOT | 1-SHOT | 3-SHOT | 5-SHOT | 1-SHOT | 3-SHOT | 5-SHOT |
| MLP | 36.53 | 41.40 | 44.05 | 8.63 | 13.75 | 16.64 | 49.96 | 59.00 | 67.10 | 37.66 | 43.82 | 49.44 |
| GCN | 52.60 | 57.8 | 64.52 | 19.97 | 29.89 | 34.82 | 69.14 | 85.02 | 91.66 | 55.95 | 69.38 | 73.90 |
| GAT | 46.60 | 64.08 | **75.10** | 23.68 | 33.51 | 38.63 | 66.14 | 84.05 | 88.67 | 52.73 | 72.24 | 73.75 |
| GIN | 31.28 | 40.30 | 42.20 | 12.80 | 13.75 | 14.52 | 76.16 | 88.90 | 91.22 | 66.81 | 77.61 | 79.43 |
| DGI | 40.07 | 46.73 | 50.67 | 11.75 | 15.06 | 18.24 | 70.93 | 72.37 | 85.47 | 59.37 | 63.41 | 66.68 |
| BGRL | - | 48.43 | - | - | 17.98 | - | - | 67.23 | - | - | 29.24 | - |
| GRAPHMAE | - | 49.24 | - | - | 19.12 | - | - | 69.75 | - | - | 32.07 | - |
| GIANT | - | 54.33 | - | - | 20.12 | - | - | 77.21 | - | - | 52.63 | - |
| PRODIGY | 48.23 | 58.64 | 61.09 | 21.44 | 23.69 | 25.51 | 66.10 | 79.61 | 84.30 | 54.30 | 59.58 | 62.03 |
| OFA | 52.80 | 58.68 | 59.92 | 21.34 | 22.13 | 24.01 | 83.46 | 83.14 | 83.64 | 63.48 | 65.76 | 66.51 |
| GFT | **58.20** | 66.00 | 68.00 | 26.49 | 34.36 | 36.29 | 88.07 | 88.53 | 89.13 | 74.97 | 74.56 | 75.01 |
| RWPT | 52.76 | **72.06** | 73.58 | **26.72** | **39.72** | **43.14** | **93.24** | **94.34** | **95.28** | **82.45** | **88.85** | **90.67** |

## G   PROMPT TEMPLATES FOR FEATURE EXTRACTION

We use Llama2-7b to extract input features for RWPT from node/edge/dataset descriptions. These descriptions come from two primary sources: (1) websites cited in the study of text-attributed graphs (He et al., 2023b; Chen et al., 2023b) and (2) hand-crafted text outlined in OFA (Liu et al., 2024a). Details on the sources of text attributes are provided in Section E. The text prompts for feature extraction in each dataset are summarized in Table 7.

## H   MORE RESULTS IN ABLATION STUDY

### H.1   RANDOM WALK VS NEIGHBORHOOD SAMPLING

We compare two sequence inputs in Section 5.4. The results are reported in Table 8.

In RWPT, the computational complexity of the attention mechanism is $\mathcal{O}(k \cdot l^2)$ rather than the standard $\mathcal{O}((kl)^2)$. This is achieved through our per-walk attention masking (Section 3.3), which restricts interactions primarily to within individual walks. This linear scaling with $k$ allows us to increase the number of walks to enhance structural coverage without incurring a quadratic computational cost. Theoretically, as detailed in Theorem 4.4, a sufficient $k$ paired with $l$ enables the reconstruction of local neighborhoods with high probability, implying that increasing $k$ should generally improve performance up to the computational budget. This is empirically confirmed by Table 8, where $(k = 8, l = 4)$ outperforms $(k = 4, l = 4)$ and $(k = 8, l = 8)$ outperforms $(k = 4, l = 8)$ on average. While a larger $l$ benefits graphs with long-range dependencies (e.g., Peptides-func and Peptides-struct), a modest choice $l \in [2, 8]$ shows robustness for the datasets considered.

### H.2   ABLATION ON DIFFERENT LLMS

We compare four LLMs: DistilBERT (Sanh et al., 2019), E5-large-v2 (Wang et al., 2022), Qwen3-Embedding-0.6B (Zhang et al., 2025), and Llama2-7B (Touvron et al., 2023) and analyze the impact of different text encoders. The results are shown in Table 9.

### H.3   CONTEXTUAL LOSS

We evaluate the proposed contextual loss (9) against a few alternatives, including other forms of context prediction and the possible inclusion of reconstruction losses. For ease of comparison, we repeat (9) here:

$$\mathcal{L}_{sample} = -\frac{1}{\ell} \sum_{j=1}^{\ell} \left( \log \mathrm{MLP}(\mathbf{h}_0 \odot \mathbf{h}_{ctx}^{(j)}) + \sum_{\forall other} \log(1 - \mathrm{MLP}(\mathbf{h}_0 \odot \mathbf{h}_{other}^{(j)})) \right),$$

Table 7: Prompting text for feature extraction.

| | | |
|---|---|---|
| CORA | NODE | "PAPER TITLE AND ABSTRACT: " + {NODE DESC} |
| | EDGE | N/A |
| | DATASET | "NODE CLASSIFICATION ON PAPER'S CATEGORY IN COMPUTER SCIENCE." |
| CITESEER | NODE | "PAPER TITLE AND ABSTRACT: " + {NODE DESC} |
| | EDGE | N/A |
| | DATASET | "NODE CLASSIFICATION ON PAPER'S CATEGORY FROM CITESEER DATASET; " + "A CITATION NETWORK OF SCIENTIFIC PAPERS." |
| PUBMED | NODE | "PAPER TITLE AND ABSTRACT: " + {NODE DESC} |
| | EDGE | N/A |
| | DATASET | "NODE CLASSIFICATION ON PAPER'S CATEGORY IN BIOMEDICAL DOMAIN." |
| ARXIV | NODE | "PAPER TITLE AND ABSTRACT: " + {NODE NAME} + " : " + {NODE DESC} |
| | EDGE | N/A |
| | DATASET | "NODE CLASSIFICATION OF LITERATURE CATEGORY COLLECTED FROM ARXIV PLATFORM." |
| WIKICS | NODE | "WIKIPEDIA ENTRY NAME: " + {NODE NAME} + ". ENTRY CONTENT: " + {NODE DESC} |
| | EDGE | N/A |
| | DATASET | "NODE CLASSIFICATION OF WIKIPEDIA ENTRY CATEGORY." |
| PRODUCTS | NODE | "PRODUCT: " + {NODE NAME} + ". DESCRIPTION: " + {NODE DESC} |
| | EDGE | N/A |
| | DATASET | "NODE CLASSIFICATION ON PRODUCT'S CATEGORY COLLECTED " + "FROM AN AMAZON PRODUCT CO-PURCHASING NETWORK." |
| WN18RR | NODE | "ENTITY AND ENTITY DESCRIPTION: " + {NODE DESC} |
| | EDGE | "RELATION BETWEEN TWO ENTITIES: " + {EDGE DESC} "RELATION BETWEEN TWO ENTITIES. THE INVERSE RELATION OF " + {EDGE DESC} |
| | DATASET | "RELATION TYPE PREDICTION BETWEEN THE CONNECTED ENTITIES OF A SUBSET OF WORDNET." |
| FB15K237 | NODE | "ENTITY NAME: " + {NODE NAME} + ", ENTITY ALTERNATIVES: " + {NODE NAME 2} + ". ENTITY DESCRIPTIONS: " + {NODE DESC} |
| | EDGE | "RELATION BETWEEN TWO ENTITIES: " + {EDGE DESC} "RELATION BETWEEN TWO ENTITIES. THE INVERSE RELATION OF " + {EDGE DESC} |
| | DATASET | "RELATION TYPE PREDICTION BETWEEN THE CONNECTED ENTITIES OF FREEBASE ENTITY PAIRS." |
| HIV | NODE | {NODE/ATOM NAME & PROPERTIES} |
| | EDGE | {EDGE/CHEMICAL BOND NAME & PROPERTIES} |
| | DATASET | "GRAPH CLASSIFICATION ON MOLECULE PROPERTY ON DATASET: HIV." |
| PCBA | NODE | {NODE/ATOM NAME & PROPERTIES} |
| | EDGE | {EDGE/CHEMICAL BOND NAME & PROPERTIES} |
| | DATASET | "GRAPH CLASSIFICATION ON MOLECULE PROPERTY ON DATASET: PCBA." |
| CHEMBL | NODE | {NODE/ATOM NAME & PROPERTIES} |
| | EDGE | {EDGE/CHEMICAL BOND NAME & PROPERTIES} |
| | DATASET | "GRAPH CLASSIFICATION ON MOLECULE PROPERTY ON DATASET: CHEMBL." |
| TOX21 | NODE | {NODE/ATOM NAME & PROPERTIES} |
| | EDGE | {EDGE/CHEMICAL BOND NAME & PROPERTIES} |
| | DATASET | "GRAPH CLASSIFICATION ON MOLECULE PROPERTY ON DATASET: TOX21." |
| PEPTIDES-FUNC | NODE | {NODE/ATOM NAME & PROPERTIES} |
| | EDGE | {EDGE/CHEMICAL BOND NAME & PROPERTIES} |
| | DATASET | "GRAPH CLASSIFICATION ON MOLECULE PROPERTY ON DATASET: PEPTIDES-FUNC." |
| PEPTIDES-STRUCT | NODE | {NODE/ATOM NAME & PROPERTIES} |
| | EDGE | {EDGE/CHEMICAL BOND NAME & PROPERTIES} |
| | DATASET | "GRAPH CLASSIFICATION ON MOLECULE PROPERTY ON DATASET: PEPTIDES-STRUCT." |
| ELLIPTIC-BITCOIN | NODE | NODE ID: {NODE ID}. FEATURE 0 {X[0]}, FEATURE 1 {X[1]}, ... |
| | EDGE | N/A |
| | DATASET | "NODE CLASSIFICATION OF THE ELLIPTIC BITCOIN DATASET OF BITCOIN TRANSACTIONS CATEGORY." |

recalling that $\mathbf{h}_0$ is the node representation computed by RWPT, $\mathbf{h}_{ctx}^{(j)}$ is its $j$-hop context, and $\mathbf{h}_{other}^{(j)}$ is the $j$-hop context of other nodes in the training batch. Experiment results are summarized in Tables 10 and 11; they are discussed in the main text (Section 5.4).

Several variants of the contrastive loss that performs context prediction exist in the graph literature. We consider DGI (Veličković et al., 2019), GraphPrompt (Liu et al., 2023b), and MaskGAE (Li et al., 2023) and adapt their losses for sequence models. For clarity, we use $\mathbf{h}_{ctx}^{(j),+}$ to denote the context for the concerned node; $\mathbf{h}_{ctx}^{(j),-}$ to denote the negative context; and when positive and negative are not distinguished, we use the notation $\mathbf{h}_{ctx}^{(j),i}$. In the latter notation, $i$ ranges over all nodes in the batch.

**DGI.** In this design, each node is associated with two sequences: one generated by the random walk sampler and the other by random nodes. The first sequence computes a positive context $\mathbf{h}_{ctx}^{(j),+}$ and is assigned label 1; while the other computes a negative context $\mathbf{h}_{ctx}^{(j),-}$ and is assigned label 0. The

Table 8: Performance comparison between random walk sampler and neighbor sampler. For Peptides-struct, the lower the better; while for others, the higher the better.

| | $k$ | $\ell$ | PUBMED | WIKICS | ARXIV | WN18RR | FB15K237 | HIV | TOX21 | PEP-F | PEP-S |
|---|---|---|---|---|---|---|---|---|---|---|---|
| WALK | 16 | 2 | **76.75** | **80.80** | 75.15 | 94.32 | **95.06** | 75.24 | **74.38** | 88.31 | 0.2599 |
| | 4 | 4 | 76.32 | 79.51 | 74.94 | 94.95 | 94.78 | 75.78 | 73.23 | 86.91 | 0.2703 |
| | 8 | 4 | 73.03 | 80.45 | **75.16** | 95.20 | 94.95 | 74.03 | 74.16 | 88.21 | 0.2582 |
| | 4 | 8 | 75.20 | 79.74 | 74.71 | 94.50 | 94.87 | 70.70 | 73.93 | 88.57 | **0.2561** |
| | 8 | 8 | 75.79 | 80.25 | 74.87 | 94.78 | 94.86 | 75.14 | 73.81 | **90.48** | 0.2574 |
| NEIGHBOR | [4, 4, 2] | | 74.23 | 80.46 | 75.15 | **95.55** | 95.05 | 71.78 | 72.40 | 87.58 | 0.2617 |
| | [8, 4] | | 71.40 | 80.02 | 75.05 | 94.95 | 95.02 | 72.81 | 73.13 | 87.37 | 0.2625 |

Table 9: Performance comparison between different LLMs for pre-processing textual features of graphs.

| LLM | DIM. | PUBMED | WIKICS | ARXIV | WN18RR | FB15K237 | HIV | TOX21 |
|---|---|---|---|---|---|---|---|---|
| DISTILBERT | 768 | 65.54 | 72.38 | 69.52 | 85.74 | 93.31 | 71.77 | 73.88 |
| E5-LARGE-V2 | 1,024 | 79.41 | 75.68 | 72.25 | 93.11 | **94.63** | 72.39 | **74.73** |
| QWEN3-EMBEDDING-0.6B | 1,024 | **81.31** | 76.76 | 72.83 | 91.93 | 94.18 | 71.95 | 73.18 |
| LLAMA2-7B | 4,096 | 76.26 | **80.27** | **75.14** | **95.25** | 94.47 | **75.15** | 74.49 |

task is to predict the correct context and hence the loss is the binary cross-entropy:

$$\mathcal{L}_{sample} = -\frac{1}{2\ell} \sum_{j=1}^{\ell} \Big( \log g(\mathbf{h}_0, \mathbf{h}_{ctx}^{(j),+}) + \log(1 - g(\mathbf{h}_0, \mathbf{h}_{ctx}^{(j),-})) \Big), \tag{23}$$

where $g$ is the Sigmoid function applied to the inner product of two representations.

**GraphPrompt.** In this design, the loss is the log-softmax of the similarity between the representations of the node and all the contexts in the training batch (one positive and all others negative):

$$\mathcal{L}_{sample} = -\frac{1}{\ell} \sum_{j=1}^{\ell} \log \frac{\exp(\text{sim}(\mathbf{h}_0, \mathbf{h}_{ctx}^{(j),i})/\tau)}{\sum_i \exp(\text{sim}(\mathbf{h}_0, \mathbf{h}_{ctx}^{(j),i})/\tau)}, \tag{24}$$

where `sim` denotes cosine similarity.

**MaskGAE.** The original MaskGAE loss contrasts positive and negative edges. We adapt this idea by predicting the $j$-hop context for the $(j-1)$-hop context, rather than for the root node:

$$\mathcal{L}_{sample} = -\frac{1}{\ell} \sum_{j=1}^{\ell} \Big( \log g(\mathbf{h}_{ctx}^{(j-1)}, \mathbf{h}_{ctx}^{(j),+}) + \sum_{\forall \mathbf{h}_{ctx}^{(j),-}} \log(1 - g(\mathbf{h}_{ctx}^{(j-1)}, \mathbf{h}_{ctx}^{(j),-})) \Big), \tag{25}$$

where the zero-hop context is simply the root node itself: $\mathbf{h}_{ctx}^{(0)} := \mathbf{h}_0$. Similar to DGI, $g$ is the Sigmoid function applied to the inner product of two representations.

In Table 10, we observe that the proposed contextual loss performs better than the three alternatives.

H.4 RECONSTRUCTION LOSS

In addition to context prediction, we consider reconstruction, which is common in the pre-training of LLM encoders, such as the mask-token prediction in BERT (Kenton & Toutanova, 2019). In this setup, one randomly masks 15% of the nodes in the input sequence and replaces them with a special token. Then, the reconstruction is to have RWPT predict these tokens in the output.

We compare three configurations: (1) no reconstruction loss (i.e., context prediction only); (2) adding the reconstruction loss for token embeddings of randomly masked nodes; and (3) adding the reconstruction loss for positional embeddings of randomly masked nodes. In Table 11, we observe that adding a reconstruction loss causes marginal differences. Hence, for cost reasons, we do not use a reconstruction loss.

Table 10: Performance comparison between contextual losses.

| | CORA | CITESEER | PUBMED | WIKICS | ARXIV |
|---|---|---|---|---|---|
| OUR LOSS (9) | **78.02 ± 0.37** | **75.46 ± 0.45** | 76.26 ± 0.53 | 80.27 ± 0.19 | **75.14 ± 0.15** |
| DGI | 76.78 ± 0.61 | 74.37 ± 0.88 | 76.99 ± 0.17 | 80.34 ± 0.26 | 75.13 ± 0.12 |
| GRAPHPROMPT | 74.97 ± 0.35 | 74.45 ± 0.60 | **77.86 ± 0.19** | 80.01 ± 0.29 | 75.08 ± 0.20 |
| MASKGAE | 77.51 ± 0.28 | 74.87 ± 0.43 | 74.81 ± 0.30 | **80.48 ± 0.13** | 75.05 ± 0.10 |

| | WN18RR | FB15K237 | HIV | TOX21 |
|---|---|---|---|---|
| OUR LOSS (9) | **95.25 ± 0.00** | 94.47 ± 0.06 | **75.15 ± 2.39** | 74.49 ± 1.40 |
| DGI | 94.55 ± 0.32 | 94.71 ± 0.11 | 72.74 ± 1.30 | 73.30 ± 0.43 |
| GRAPHPROMPT | 94.28 ± 0.07 | **94.89 ± 0.08** | 73.46 ± 2.40 | **75.55 ± 0.54** |
| MASKGAE | 94.64 ± 0.13 | 94.79 ± 0.10 | 74.62 ± 1.77 | 72.83 ± 1.07 |

Table 11: Performance comparison between training losses.

| | CORA | CITESEER | PUBMED | WIKICS | ARXIV |
|---|---|---|---|---|---|
| CONTEXT PREDICTION (9) | 78.02 ± 0.37 | **75.46 ± 0.45** | **76.26 ± 0.53** | 80.27 ± 0.19 | 75.14 ± 0.15 |
| + TOKEN RECONSTRUCTION | **78.18 ± 0.52** | 75.25 ± 0.71 | 75.65 ± 0.18 | **80.74 ± 0.34** | **75.33 ± 0.19** |
| + POSITION RECONSTRUCTION | 76.61 ± 0.44 | 74.69 ± 0.43 | 74.59 ± 0.20 | 80.25 ± 0.23 | 75.00 ± 0.13 |

| | WN18RR | FB15K237 | HIV | TOX21 |
|---|---|---|---|---|
| CONTEXT PREDICTION (9) | **95.25 ± 0.00** | 94.47 ± 0.06 | 75.15 ± 2.39 | **74.49 ± 1.40** |
| + TOKEN RECONSTRUCTION | 94.54 ± 0.32 | **94.85 ± 0.07** | 72.77 ± 1.07 | 74.41 ± 0.48 |
| + POSITION RECONSTRUCTION | 94.38 ± 0.25 | 94.67 ± 0.16 | **75.33 ± 0.91** | 74.42 ± 0.76 |

## H.5 ARCHITECTURE DESIGN

Our architecture design follows the standard decoder-only Transformer (GPT-2 (Radford, 2018)), with two modifications: (1) using edge features in each Transformer layer (see Section 3.2); and (2) a customized attention mask (see Section 3.3). This ablation study evaluates the impact of the changes, as summarized in Table 12.

**Edge feature input.** In our model, edge features are incorporated at each transformer layer, similar to GNNs (Hu et al., 2020a), by attaching a projection head to every transformer layer. Alternatively, as in prior work, edge features can be introduced at the input level by adding tokenized edge embeddings alongside node and positional embeddings.

Using edge features only at the input level yields inconsistent effects on node- and link-level tasks–two datasets show improvements and two show degradation–due to the absence of edge features in pre-training on these datasets. Worse, it consistently harms the performance of graph-level tasks.

**Full attention mask.** Replacing our attention mask (Figure 2) with the standard full attention mask used in BERT (Kenton & Toutanova, 2019) allows each contextual node to attend to any other node sampled by other random walks. However, doing so will introduce spurious connections; for instance, nodes from two random walks moving in opposite directions–typically weakly connected–would be treated as fully connected under the full attention mask.

As a result, using full attention results in performance degradation on five out of six benchmarks. Nevertheless, the impact is moderate, as many of the introduced connections are either correct or only mildly disruptive due to all nodes residing within the same local neighborhood.

Table 12: Performance comparison on different model architectures.

| | WIKICS | ARXIV | WN18RR | FB15K237 | HIV | TOX21 |
|---|---|---|---|---|---|---|
| OUR MODEL | 80.27 ± 0.19 | **75.14 ± 0.15** | **95.25 ± 0.00** | 94.47 ± 0.06 | **75.15 ± 2.39** | **74.49 ± 1.40** |
| EDGE FEAT. INPUT | **80.93 ± 0.26** | 74.92 ± 0.20 | 94.67 ± 0.19 | **94.81 ± 0.01** | 73.15 ± 2.95 | 73.54 ± 0.27 |
| FULL ATTN. MASK | 79.55 ± 0.12 | 74.62 ± 0.21 | 94.44 ± 0.14 | 94.78 ± 0.11 | 73.42 ± 3.43 | 73.78 ± 1.08 |

## H.6 GRAPHS WITH NON-TEXTUAL FEATURES

We consider the Elliptic-Bitcoin dataset (Weber et al., 2019) as an example for handling non-textual graphs. We fine-tune the prediction head on the training subset of the graph and evaluate it on the test subset. This experiment is noteworthy for two reasons: (1) the pre-training stage does not involve any financial-domain data, and (2) the fine-tuning procedure does not use textual features. To align raw features with the Transformer's input dimension, we consider three simple adaptations: (i) random projection, (ii) repetition-based padding, and (iii) an artificial prompt construction for raw features (see Table 7). Because the dataset is highly imbalanced, we follow Weber et al. (2019) and report precision, recall, and F1 for the minority class. Results are shown in Table 13.

Overall, our RWPT model achieves superior performance by using projection or padding, while falls behind simple baselines by using the artificial prompt. The baselines do not generalize well to certain feature types; e.g., bag-of-words, since such raw features can already be viewed as a primitive language model, which is against one of the key assumptions of our framework: that stronger LLM representations are beneficial.

Table 13: Performance on Elliptic Bitcoin dataset, which has non-textual features only.

| METRIC | PRECISION | RECALL | F1-SCORE |
|---|---|---|---|
| MLP | 66.80 ± 6.11 | 61.61 ± 1.32 | 64.07 ± 2.94 |
| GCN | 79.98 ± 1.68 | 47.29 ± 1.81 | 59.45 ± 1.59 |
| RWPT W/ PROJECTION | 87.26 ± 2.79 | **63.02 ± 0.93** | **73.15 ± 0.62** |
| RWPT W/ PADDING | **92.58 ± 0.75** | 56.96 ± 0.94 | 70.52 ± 0.68 |
| RWPT W/ DESCRIPTION | 74.56 ± 3.37 | 52.88 ± 2.07 | 61.80 ± 0.28 |

## H.7 ATTENTION SCORES

In this section, we present attention score visualizations during inference to verify that our model approach effectively learns random walk patterns that enable domain transfer rather than relying on simple heuristics such as averaging.

We consider the model pre-trained solely on the arXiv dataset and randomly sample nodes from the WN18RR dataset for inference. We extract attention scores from randomly selected nodes, layers, and heads. From Figure 6, we observe a clear block-diagonal structure, as expected in Figure 2; each block corresponds to a walk segment and exhibits varied attention patterns. Notably, the first two columns (dataset virtual node and root node) consistently receive higher attention scores because both are key information that enable batching with cross-domain datasets simultaneously.

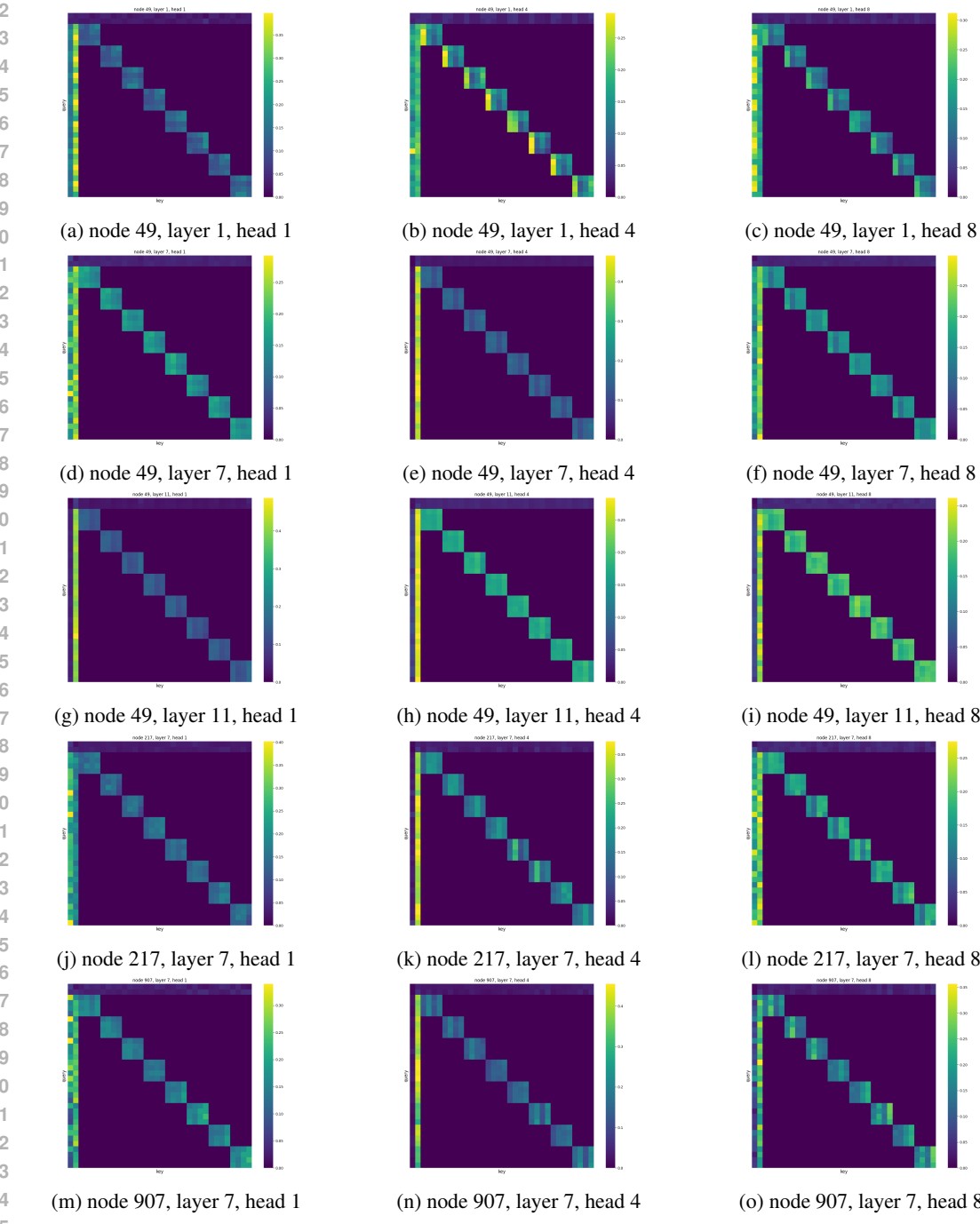

Figure 6: Attention scores of different sampled nodes, layers, and heads.

# I  LLMS USAGE DISCLOSURE

Besides the pre-trained LLMs used in data pre-processing step as described in the method section, LLMs were used in this work only for limited, non-scientific purposes. Specifically, GPT-5 was employed to assist with grammar correction and word choice during the preparation of the manuscript. All literature survey, research ideation, model design, experiments, and analysis were conducted solely by the authors without generative assistance.

