# OpenReview forum: "Cross-Domain Pre-training of Transformers on Text-Attributed Graphs via Random Walks"
_ICLR.cc/2026/Conference — Submitted to ICLR 2026_

### Official Review · Reviewer_nDuC · 2025-10-30

**Soundness:** 4
**Presentation:** 2
**Contribution:** 3
**Rating:** 6
**Confidence:** 2

**Summary:**

This paper proposes a unified Transformer-based framework for cross-domain graph pre-training, focusing on text-attributed graphs. The key idea is to represent graph structures as sequential inputs via random walks, enabling the use of Transformer architectures similar to those in NLP.The paper outlines four desiderata for pre-training across diverse graphs and demonstrates strong transferability in out-of-domain evaluation.

**Strengths:**

Originality：Introduces a new paradigm for cross-domain graph pre-training, using random walks and LLM-based embeddings to unify diverse graph structures；3.1, the dataset-level virtual tokens to distinguish graphs is creative.

Quality : Comprehensive experimental validation across multiple domains demonstrates strong and consistent performance;The results show good transferability even when pre-training on only a few representative datasets, confirming the framework’s generalization ability.

Clarify：.
Significance：Demonstrates that large-scale pre-training on a few datasets can generalize across domains, a valuable insight for developing graph foundation models

Significance：Demonstrates that large-scale pre-training on a few datasets can generalize across domains, a valuable insight for developing graph foundation models

**Weaknesses:**

While the paper discusses edge incorporation, more analysis on label leakage and task-specific fine-tuning would strengthen the claims.
The ablation on walk length ℓ and number of walks k could include a more detailed complexity discussion.

**Questions:**

How does the model perform when the node text information is noisy or incomplete?

How would the framework adapt to temporal or dynamic graphs?

---

> ### Author Response · Authors · 2025-11-25
>
> We thank the reviewer for recognizing the novelty and significance of our work. We are pleased that you appreciate our proposed RWPT framework, particularly the creative use of random walks and dataset-level virtual tokens to enable a unified Transformer architecture for diverse graphs, and the potential of our approach for developing graph foundation models. We address your specific questions and concerns below.
>
> > Q1. How does the model perform when the node text information is noisy or incomplete?
>
> For text-noisy or incomplete graphs, RWPT's performance depends on the interplay between text quality and structural information. Our experiments on knowledge graphs (WN18RR, FB15k237) demonstrate this well: although only entity names and tags are available, RWPT achieves comparable performance by effectively leveraging graph structure. This shows the model still functions well when text is sparse. Table 13 (Section H.6) presents another experiment on the Elliptic-Bitcoin dataset, a financial network containing only raw numerical features without explicit textual descriptions. Here, RWPT employs simple feature alignment strategies (projection and padding) to transform numerical features to the required input dimension. The results show that the framework generalizes to non-textual graphs in some cases.
>
>
> > Q2. How would the framework adapt to temporal or dynamic graphs?
>
> Our framework extends naturally to temporal graphs by constraining random walk sampling to respect temporal edge orderings. However, the contrastive learning objective would require modifications to handle evolving graph structure and temporal information. When the graph information is relatively "complete" at each snapshot (that is, the graph does not just contain one or two dangling edges emerging during the snapshot, but all existing edges at the time), one may just treat each snapshot a separate graph and perform the usual random walk sampling. Edge and/or node features may change over time but they are attached to the random walks for the snapshot.
>
> > W1. While the paper discusses edge incorporation, more analysis on label leakage and task-specific fine-tuning would strengthen the claims. The ablation on walk length ℓ and number of walks k could include a more detailed complexity discussion.
>
> **On Label Leakage:** We want to clarify that RWPT already incorporates explicit mechanisms to prevent label leakage. Edge features are only used for graph-level tasks (Section 3.2, line 215), not for node- or link-level tasks. This design choice is intentional because edge attributes can directly leak the link-prediction target on link-level tasks. For graph-level tasks, edges are incorporated at every Transformer block via independent projection heads (similar to the design in OGB baselines [(Hu et al., 2020)](https://arxiv.org/abs/2005.00687)). Additionally, our contrastive learning objective (Eq. 9) maximizes mutual information between root nodes and their random walk contexts while minimizing it between different root nodes. This self-supervised objective never accesses downstream task labels during pre-training.
>
> **On Complexity Discussion:** We have added a more detailed discussion on choices of k and l and how they affect the complexity as well as the performance. This discussion can be found in Section H.1 in the revised paper.

---

### Official Review · Reviewer_gSVb · 2025-10-31

**Soundness:** 3
**Presentation:** 3
**Contribution:** 3
**Rating:** 6
**Confidence:** 4

**Summary:**

The paper presents a Transformer-based model for text-attributed graph representation learning, utilizing random walks to represent graph nodes as sequences. Through a custom context prediction loss for self-supervised pre-training, the approach can handle the graph types from small molecules to large citation networks, and can be easily adapted to various downstream tasks via fine-tuning.

**Strengths:**

1. The paper proposes a single yet effective approach by transforming nodes into random walk sequences that can be processed by a Transformer, making it adaptable to diverse graph types.
2. The model leverages self-supervised pre-training to adapt to diverse datasets for cross-domain learning, offering significant transferability across tasks.
3. The paper provides a theoretical analysis of random walks’ expressiveness.

**Weaknesses:**

1. Lacking comparisons with recent studies, such as LLM-based methods like LLaGA  (Chen et al., 2024b), GraphGPT  (Tang et al., 2024), and unified GNN models like UniGraph  (He & Hooi, 2024), which are mentioned in Appendix A.
2. The figures are kind of hard to interpret without accompanying figure captions or explanations.

**Questions:**

1. In the few-shot learning results shown in Table 6, the proposed method achieves strong performance across nearly all settings, but it underperforms on the ARXIV 5-way task under both 1-shot and 5-shot conditions. Could the authors provide insights into the possible reasons for this?
2. Molecular graphs and knowledge graphs differ significantly in scales. Could the authors discuss some theoretical or empirical guidelines for selecting the walk length $l$ and the number of walks $k$ for these different graph types?
3. Why can representing molecular graph structures through random walks achieve performance comparable to graph transformer models such as GPS, which incorporate structural encodings? Could the authors provide more analysis?

---

> ### Author Response · Authors · 2025-11-25
>
> We appreciate the reviewer for acknowledging the effectiveness of our unified random walk-based framework in handling diverse graph domains transfer, and providing theoretical guarantees of the expressivity. We address your specific questions below.
>
> > Q1. In the few-shot learning results shown in Table 6, the proposed method achieves strong performance across nearly all settings, but it underperforms on the ARXIV 5-way task under both 1-shot and 5-shot conditions. Could the authors provide insights into the possible reasons for this?
>
> We believe that the underperformance on the arXiv 5-way 1-shot and 5-way 5-shot tasks comes from a deliberate trade-off in our lightweight downstream head choice (MLP with cosine similarity), which we intentionally chose to highlight our framework's contribution rather than task-specific tuning.
> arXiv's high intra-class variance requires strong immediate discrimination in few-shot tasks, yet our model remains competitive with GCN at 1-shot (52.76% vs 52.60%) and ranks second at 5-shot (73.58% vs 75.10% for GAT). The fast improvement from 1-shot to 5-shot (52.76% → 73.58%) confirms our pre-trained representations are high-quality.
>
> > Q2. Molecular graphs and knowledge graphs differ significantly in scales. Could the authors discuss some theoretical or empirical guidelines for selecting the walk length and the number of walks for these different graph types?
>
> Empirically, we find that larger $k$ generally yields better performance up to computational budget limits. This aligns with our theoretical analysis (Theorem 4.4) which shows $k = \max(nl, n^2/l^2)$ is required for ego-graph reconstruction with radius $r = l$. This bound is typically larger than the actual $k$ used in practice. In our experiments, $k=8$ with $l=4$ works well across all datasets. On the other hand, long-range graphs (Peptides-func, Peptides-struct in Table 8) suggest to use larger $l$ for graphs where long-range dependencies matter; otherwise, $l \in [4,8]$ provides robust performance. The key insight is that, it's more important to match node representation requirements rather than the scale of the graph.
>
> > Q3. Why can representing molecular graph structures through random walks achieve performance comparable to graph transformer models such as GPS, which incorporate structural encodings? Could the authors provide more analysis?
>
> We have theoretically proved that ego-graphs can be reconstructed from our single-source random walks combined with shortest-path distance positional encodings. On molecular graphs, the bounded size means that a modest number of random walks suffices to reliably sample the neighborhood. Since each ego-graph in molecular graphs represents a substantial portion of the entire graph, this makes the method both theoretically and empirically efficient for implicitly reconstructing the graph and performing graph-level tasks.
>
> > W1. Lacking comparisons with recent studies, such as LLM-based methods like LLaGA (Chen et al., 2024b), GraphGPT (Tang et al., 2024), and unified GNN models like UniGraph (He & Hooi, 2024), which are mentioned in Appendix A.
>
> We appreciate the reviewer's suggestion to include recent LLM-based models and unified GNN models. We respectfully highlight that Table 1 already includes representative baselines from these categories: FUG [(Zhao et al., 2024)](https://proceedings.neurips.cc/paper_files/paper/2024/hash/075b7d4bd7fc32d9cf468a7b67c38d15-Abstract-Conference.html) as a state-of-the-art unified GNN and GraphCLIP [(Zhu et al., 2025)](https://dl.acm.org/doi/abs/10.1145/3696410.3714801?casa_token=vwo7HVr8whcAAAAA:X713h5dr3ptYLyTbAxLXkjmIqzzjiYUKgh7zqv3u7ZpLO0ngr8ye17mKR8ShQS0JBntC_hp8RSSL) as a representative LLM-based method. Regarding the other specific methods mentioned (UniGraph, GraphGPT, LLaGA), a fair comparison is currently infeasible due to reproducibility hurdles: the official implementation of UniGraph (He et al., 2024) has incomplete code (see GitHub [Issue #3](https://github.com/yf-he/UniGraph/issues/3)), while GraphGPT (Tang et al., 2024) and LLaGA (Chen et al., 2024b) lack official instruction-tuning templates for the specific Knowledge Graph (WN18RR, FB15k237) and Molecular (HIV, PCBA, Tox21) datasets used in our study.
>
> > W2. The figures are kind of hard to interpret without accompanying figure captions or explanations.
>
> We appreciate the feedback on figure readability. In the revised version, we have added detailed captions to Figures 2 and 3 that explicitly describe the key insights being illustrated.

---

### Official Review · Reviewer_MqTi · 2025-11-01

**Soundness:** 2
**Presentation:** 3
**Contribution:** 2
**Rating:** 4
**Confidence:** 2

**Summary:**

The paper introduces RWPT, a cross-domain graph pre-training approach that encodes each node using multiple random walks and processes them with a Transformer, trained via a contrastive context prediction loss. The model transfers well across datasets and tasks.

**Strengths:**

1. Effective cross-domain transfer without retraining the backbone.
2. Random-walk representation captures long-range structure.
3. Strong results across node, link, and graph classification.

**Weaknesses:**

1. The topic has been widely explored before, lack novelty.
2. Computationally heavy due to long sequences and Transformers.
3. Relies on text-attributed nodes; less applicable to non-text graphs.

**Questions:**

See weaknesses.

---

> ### Author Response · Authors · 2025-11-25
>
> We thank the reviewer for recognizing our contribution in enabling efficient cross-domain transfer without backbone retraining and the capability of our random-walk representation to capture long-range structural dependencies. Below, we address your concerns and questions point-by-point.
>
> > W1. The topic has been widely explored before, lack novelty.
>
> While graph pre-trained models and random walks for representation learning are gaining increasing traction, our contribution lies in a novel systematic design for cross-domain pre-training. RWPT has three unique elements: (1) the biased random walks enabling flexible handling of graphs of various scales with predictable batching constraints; (2) the shortest-path distance positional encoding with theoretical analysis (Theorems 4.4–4.5) guaranteeing GD-WL expressivity of our efficient random walk sampler; and (3) the contrastive context loss that outperforms alternative training objectives (Tables 10–11). The ablation studies (Section 5.4, Section H) demonstrate that each component is essential: random walks alone lack structural awareness, LLM features alone lack graph topology awareness, and standard Transformers are unsuitable for structured data.
> We also proposed four desiderata in the Introduction for building a cross-domain pre-trained graph model and our framework addresses all of them.
>
> > W2. Computationally heavy due to long sequences and Transformers.
>
> We appreciate the concern about computational cost. While pre-training is expensive as a one-time cost, inference is efficient by design. Firstly, our per-walk attention mask (Section 3.3, Figure 2) reduces complexity by a factor of k, the number of walks. We implemented a loop-based kernel that processes one block and prefix per iteration, achieving a 15% speedup on ArXiv compared to PyTorch's built-in Flash Attention. Secondly, to support the complexity analysis in Section D, empirical results on PubMed shows that: random walk sampling costs 36ms, RWPT inference 19s, and task-head training 40ms per epoch. The efficiency gains largely come from cross-domain transfer, avoiding the need to train separate models per dataset. Using just three representative datasets for pre-training (Table 2) achieves performance close to using all ten, demonstrating strong transferability and favorable total cost amortization across multiple downstream tasks.
>
> > W3. Relies on text-attributed nodes; less applicable to non-text graphs.
>
> RWPT is intended for text-attributed graphs, as stated in the title and core methodology. Nevertheless, the framework can generalize to non-textural graphs in some cases, particularly when the graph structure is crucial to the task. Table 13 (Section H.6) presents an example on the Elliptic-Bitcoin dataset, a financial network containing only raw numerical features without explicit textual descriptions. Here, RWPT employs simple feature alignment strategies (projection and padding) to transform numerical features to the required input dimension. The results show that the framework generalizes to non-textual graphs in this case. For more general cases, one may explore using trainable adapters to handle numerical features or prompt templates to solicit text, but the exploration is out of the scope here.

---

### Official Review · Reviewer_199r · 2025-11-02

**Soundness:** 2
**Presentation:** 2
**Contribution:** 1
**Rating:** 2
**Confidence:** 5

**Summary:**

This work explores pre-training graph representations using Transformer architectures. It represents nodes through collections of random walks, enabling Transformers to model diverse graph structures. The study provides theoretical justification for this representation and introduces a random-walk-based context prediction loss, demonstrating strong transferability.

**Strengths:**

1. The paper is well-written and easy to follow.
2. The idea of sequentializing graphs with random walks is reasonable.

**Weaknesses:**

1. The proposed method requires positional encoding based on shortest-path distances, which explicitly injects the graph’s inductive bias into the model. As a result, the reconstruction of local neighborhoods is unsurprising. In practice, computing shortest-path distances for large graphs can be time-consuming, limiting the method’s scalability.

2. The proposed method claims to capture long-range dependencies; however, in the main experiments, the walk length was set to 4, which is insufficient to support such a claim. With 8×4 short walks, the sampled contexts likely resemble local k-hop neighborhoods, suggesting that the properties of random walks are not fully exploited.

3. The method involves the use of advanced LLMs (e.g., LLaMA), which produce high-quality node features initially. For a fair comparison, the authors should clearly specify the feature types used in the baselines, as the observed improvements may stem from the strong textual embeddings rather than the proposed framework itself.

4. Many experimental details are missing, such as dataset splits. Table 1 shows that pretrained graph models perform consistently better than individually trained baselines, which contradicts prior findings that graph pretraining often fails to yield significant gains.

5. The transfer learning results in Table 2 are surprisingly strong, e.g., pretraining on a single out-of-distribution dataset leads to substantial improvements over training from scratch. This is counter-intuitive and arguably too good to be true, yet the authors provide no explanation for it.

6. Table 2 and Fig. 4 show that RWPT achieves significantly larger gains on link- and graph-level tasks than on node-level tasks. The authors should clarify the factors contributing to this discrepancy.

7. Following 5, a Transformer pretrained on limited (and potentially OOD) data is unlikely to learn complex random walk patterns. Given the strong results, the model may be relying on simple heuristics, such as mean aggregation. To verify that the model has learned meaningful interaction patterns, the authors are encouraged to analyze intermediate representations (e.g., attention patterns). Additionally, a simple baseline using non-parametric mean aggregation over LLM features, followed by an MLP task head (optionally with shortest-path encodings), should be included to provide a fair comparison and establish a starting point for evaluating more complex methods such as RWPT.

**Questions:**

See Weaknesses.

---

> ### Author Response · Authors · 2025-11-25
>
> We thank the reviewer for their constructive feedback and for recognizing our core methodology of sequentializing graphs via random walks is reasonable and effective. We address your specific concerns below.
> > W1
>
> **On Scalability:** We only compute single-source shortest-path distances locally. Our complexity is $O(k \cdot l \cdot d)$, where $k$ is the number of walks, $l$ is the walk length, and $d$ is the local degree bound. This is independent of global graph size $|V|$ and $|E|$. In practice, we provide empirical evidence in Section D which shows that random walk sampling is not a bottleneck. Regarding inductive bias, we concur with recent findings (e.g., Neural Graph Pattern Machine, Wang et al., 2025) that structural positional encodings are essential for extracting expressive power from random walks.
>
> **On Theoretical Contribution:** While the ego-graph reconstruction based on shortest-path distances may not be surprising, we would like to highlight that not all distances are needed. We quantify the needed distances in Theorem 4.4, which states that an ego-graph requires $k=O(\max(nr,n^2/r^2))$ random walks of length $r$ for reconstruction, matching the known worst-case of $O(n^2)$ SP oracles. Critically, we only employ pseudo SP oracles rather than exact SP oracles, which is very efficient.
> > W2
>
> We validate long-range capabilities in Table 8 using the Long Range Graph Benchmark (Dwivedi et al., 2022). We observe that l=8 substantially outperforms shorter walks with l=2 and 4. Our random walk sampler (with biased parameters $p=1.0, q=0.1$) behaves similarly to DFS while maintaining constant total node budget yet reaching far-distant neighbors.  Importantly, node revisiting is allowed, which is a deliberate pattern that our model learns and differs our approach from local k-hop neighborhood sampling.
> > W3
>
> To isolate the Transformer’s contribution from LLM feature quality, we compared RWPT against GCN and MLP baselines using identical LLM features. As shown in the table below, RWPT substantially outperforms baselines on link and graph tasks where text is sparse, while maintaining competitive performance on text-rich node classification. This confirms that both semantic features and structural information contribute to the effectiveness of our framework.
>
> |Method|(link) WN18RR|(link) FB15K237|(graph) HIV|(graph) Tox21|(node) Cora|(node) Pubmed|(node) WikiCS|(node) Arxiv|
> |-|-|-|-|-|-|-|-|-|
> |GCN| 82.93|86.61|73.05|69.31|**79.56**|76.22|**80.97**|74.48|
> |MLP|83.82|91.65|69.57|71.35|72.59|75.86|80.13|74.95|
> |RWPT|**95.25**|**94.47**|**75.15**|**74.49**|78.02|**76.26**|80.27|**75.14**|
> > W4
>
> **On Domain Transfer:** We appreciate the reviewer's observation, which is actually the motivation for our core design.
> Our framework is explicitly designed to mitigate negative transfer through three complementary mechanisms: (1) LLM-encoded features align diverse semantic spaces, (2) random walk sequences with SP encoding capture topology without domain-specific bias, and (3) task-specific heads allow for necessary adaptation. Therefore, our framework captures textual and structural information in a domain-agnostic manner, enabling effective transfer and task-specific adaptation which explains the improvement shown in Table 1.
>
> **On Experimental Details:** We provided experimental details in Sections E and F. Regarding dataset splits, we mirror the setup from prior work (OFA (Liu et al., 2024), GFT (Wang, 2024)). For additional datasets (Peptides, Elliptic Bitcoin), we adopt the original papers' splits.
> > W5
>
> As discussed in the above response to **W4**, our downstream experiments leverage task-specific downstream head fine-tuning (not zero-shot transfer). We deliberately use lightweight downstream heads (1–2 layer MLPs) to constrain adaptation capacity and verify the the representational power of RWPT generated embeddings.
> > W6
>
> This performance discrepancy validates our architectural choices. Text-rich node classification is largely semantic (where LLM features suffice), whereas link and graph tasks require understanding structural topology. RWPT excels in the latter because our contrastive objective (Eq. 9) forces the model to discriminate between diverse random walk patterns (from local neighborhoods to long paths) to generate structural-aware embeddings.
>
> > W7
>
> We have added visualizations of attention scores in Section H.7 to confirm our model learns complex patterns.
> For the additional experiment on the simple heuristics, the results are shown in the following table which indicate that our model has learned meaningful patterns rather than simple heuristics.
>
> | Method | (link) WN18RR | (link) FB15K237 | (graph) HIV | (graph) Tox21 | (node) Cora  | (node) Pubmed | (node) WikiCS | (node) Arxiv |
> |-|-|-|-|-|-|-|-|-|
> | mean aggr | 72.97 | 86.11 | 71.66 | 71.77 | **80.16** | 73.67 | 75.90 | 70.22 |
> | RWPT | **95.25** | **94.47** | **75.15** | **74.49** | 78.02 | **76.26** | **80.27** | **75.14** |

---

### Meta-Review · Area_Chair_gxBS · 2026-01-08

**Summary:**

Reviewers noted the following key issues:

1. Improvements may be coming from advanced LLM-based features
2. Random walk / shortest path based encoding is well explored on graphs, limiting the novelty.
3. Computational aspects lack discussion
4. Lack of comparison to recent LLM+graph models
5. Insufficient interpretation of results

**Reviewer Concerns:**

1, 5 above are largely addressed.
2 offers some clarity in the rebuttal, but not fully convincing
Response to 3 and 4 are not convincing or only partially addressed

I tend to view 2 and 4 as key hurdles for acceptance. My overall opinion is slightly leaning rejection

**Reviewer Scores:**

Probably increase by 0.5 on average.

---

### Decision · Program_Chairs · 2026-01-26

Reject